# Homology directed telomere clustering, ultrabright telomere formation and nuclear envelope rupture in cells lacking TRF2$^B$ and RAP1

Rekha Rai[1] ✉, Kevin Biju[1,2], Wenqi Sun[3,4], Tori Sodeinde[1], Amer Al-Hiyasat ®[1], Jaida Morgan[1], Xianwen Ye[4,5], Xueqing Li[3,4], Yong Chen ®[3,4,5] & Sandy Chang ®[1,6,7] ✉

Double-strand breaks (DSBs) due to genotoxic stress represent potential threats to genome stability. Dysfunctional telomeres are recognized as DSBs and are repaired by distinct DNA repair mechanisms. RAP1 and TRF2 are telomere binding proteins essential to protect telomeres from engaging in homology directed repair (HDR), but how this occurs remains unclear. In this study, we examined how the basic domain of TRF2 (TRF2$^B$) and RAP1 cooperate to repress HDR at telomeres. Telomeres lacking TRF2$^B$ and RAP1 cluster into structures termed ultrabright telomeres (UTs). HDR factors localize to UTs, and UT formation is abolished by RNaseH1, DDX21 and ADAR1p110, suggesting that they contain DNA-RNA hybrids. Interaction between the BRCT domain of RAP1 and KU70/KU80 is also required to repress UT formation. Expressing TRF2$^{ΔB}$ in $Rap1^{-/-}$ cells resulted in aberrant lamin A localization in the nuclear envelope and dramatically increased UT formation. Expressing lamin A phosphomimetic mutants induced nuclear envelope rupturing and aberrant HDR-mediated UT formation. Our results highlight the importance of shelterin and proteins in the nuclear envelope in repressing aberrant telomere-telomere recombination to maintain telomere homeostasis.

Telomeres are repetitive DNA-protein complexes that maintain genome stability through protection of linear chromosome ends from initiating DNA Damage Response (DDR) pathways and aberrant repair to maintain genome stability[1]. Telomeres in somatic cells are localized throughout the nucleus; however they are transiently enriched at the nuclear envelope during post-mitotic nuclear reassembly[2,3]. The shelterin complex, comprised of six specialized proteins, mediates telomere protection by repressing the activation of DDR pathways[4]. TRF1 (Telomeric repeat-binding factor 1) and TRF2 (Telomeric repeat-binding factor 2) bind to the double-stranded telomeric DNA, while POT1 (Protection of telomeres 1) binds to the single-stranded (ss) telomeric DNA and interacts with TPP1. TIN2 bridges TPP1-POT1 with TRF1 and TRF2. RAP1 (Repressor/activator protein 1) is a highly evolutionarily conserved protein which binds to telomeric DNA via its

[1]Department of Laboratory Medicine, Yale University School of Medicine, New Haven330 Cedar StreetCT 06520, USA. [2]Johns Hopkins University School of Medicine, Baltimore, MD 21205, USA. [3]State Key Laboratory of Molecular Biology, National Center for Protein Science Shanghai, Shanghai Institute of Biochemistry and Cell Biology, Center for Excellence in Molecular Cell Science, Chinese Academy of Sciences, Shanghai 200031, China. [4]University of Chinese Academy of Sciences, 100049 Beijing, China. [5]School of Life Science and Technology, ShanghaiTech University, 100 Haike Road, Shanghai 201210, China. [6]Department of Pathology, Yale University School of Medicine, 330 Cedar Street, New Haven, CT 06520, USA. [7]Department of Molecular Biophysics and Biochemistry, Yale University School of Medicine, 330 Cedar Street, New Haven, CT 06520, USA. ✉e-mail: rekha.rai@yale.edu; s.chang@yale.edu

interaction with TRF2[4]. TRF2 represses ATM-CHK2 signaling and C-NHEJ mediated repair by forming the t-loop structure wherein the 3′ G overhang is embedded into the double-stranded telomeric DNA[5–10]. RAP1 is not required for protection from C-NHEJ, suggesting that TRF2 alone is sufficient to repress ATM-CHK2 and C-NHEJ mediated repair[11,12]. Instead, RAP1 plays a key role in repressing aberrant HDR at telomeres without activating DDR pathways[11,12].

The N-terminal basic domain (TRF2[B]) facilitates the formation of a branched DNA structure to block the recruitment of PARP1 and branch migration by HJ resolvases to prevent the deleterious cleavage of T-loops[9,13–18]. T-loops resemble HR intermediates and, if not appropriately protected, undergo branch migration to form double Holliday junctions (dHJ), potent substrates for cleavage by HJ resolvases, including the SLX4 resolvase complex[13,19–21]. RAP1 modulates the interaction of the N-terminal basic domain (TRF2[B]) with branched DNA and prevents it from engaging in sequence-independent interactions with DNA, thus promoting the specificity of TRF2 binding to double-stranded telomeric repeats[17,22,23]. Our lab has shown previously that RAP1 cooperates with the basic domain of TRF2 (TRF2[B]) to repress PARP1 and SLX4 localization to telomeres[18]. In the absence of RAP1 and TRF2[B], PARP1 and SLX4 promote rapid telomere resection, resulting in catastrophic telomere loss and the generation of telomere-free chromosome fusions[18]. How RAP1 cooperates with TRF2[B] to repress telomere HDR is not well understood.

In this study, we examined mechanistically how TRF2[B] and RAP1 cooperate to repress telomeres clustering into structures we term ultrabright telomeres (UTs). We show that HDR factors co-localize with UTs, and that replication stress enhances their formation. Two proteins that regulate DNA-RNA hybrids, DDX21 and ADAR1p110, reduce the formation of UTs. In addition, RNaseH1 abolishes UT formation, suggesting that R-loops are present in these structures. Expressing TRF2[ΔB] in *Rap1[−/−]* cells resulted in aberrant lamin A localization in the nuclear envelope and increased telomere-telomere bridging and UT formation. Expressing lamin A phosphomimetic mutants induced nuclear envelope rupturing and aberrant HDR-mediated UT formation. Our results highlight the importance of shelterin and proteins in the nuclear envelope in repressing aberrant telomere-telomere recombination to maintain telomere homeostasis.

## Results

### TRF2[B] and RAP1 cooperate to repress ultrabright telomere formation

We have shown previously that elimination of TRF2's basic domain and deletion of its interacting partner RAP1 results in catastrophic telomere loss and telomere-free fusions[18]. To understand mechanistically how RAP1 cooperates with TRF2[B] to repress catastrophic telomere deletion, we used sh*Trf2* to remove endogenous TRF2 in *Rap1[−/−]* MEFs and then expressed vector or shRNA-resistant TRF2[ΔB] [6,11,18]. In a second experiment, we reconstituted TRF2-depleted WT MEFs with either vector, TRF2[ΔB] or the TRF2[ΔB; L286R] mutant (in human TRF2[ΔB; L288R]) that is unable to interact with endogenous RAP1[11,18]. While reconstitution of vector in the absence of endogenous TRF2 resulted in end-to-end chromosome fusions, reconstitution with TRF2[ΔB] led to rapid formation of very bright telomeric foci in ~12-15% of *Rap1[−/−]* cells that we call UTs detected by telomere-specific fluorescence in situ hybridization (PNA-FISH) (Fig. 1a, b, Supplementary Fig. 1a). UTs are usually characteristic features of the telomerase-independent telomere maintenance mechanism termed as alternative lengthening of telomeres (ALT)[24]; however, UTs formed in the absence of TRF2[B] and RAP1 are readily detected in telomerase positive cells. UTs resulted in a ~8-fold increase in average telomere foci size, resulting in saturation of the CCD detector at our standard 100 ms exposure (Supplementary Fig. 1a). UTs were not detected in WT MEFs lacking endogenous TRF2, and only rarely observed in WT cells expressing TRF2[ΔB] (Fig. 1a, b). A time course experiment revealed that UT formation peaked at ~96 h

and declined by 120 h (Fig. 1c). The decline in the number of UTs observed directly correlated with an increase in UT size (measured as area of UT focus signal, Supplementary Fig. 1b, c). Chromosome specific sub-telomere FISH revealed that UTs are formed from telomeres derived from multiple chromosomes (Fig. 1d, Supplementary Fig. 1d). The Fucci system, which utilizes the fluorescent G1 reporter CDT1 and the S/G2 reporter Geminin[25], revealed that formation of UTs occurred primarily in the S/G2 phase of the cell cycle (Fig. 1g).

Intriguingly, we also detected fine filamentous structures bridging multiple clusters of telomeres that stained brightly with a (CCCTAA)$_3$ PNA telomere probe, suggesting that these filaments are composed of telomeric DNA (Fig. 1a). Live cell imaging experiments using GFP-TRF1 to label telomeres further confirm that filamentous telomeric DNA links multiple clusters of telomeres in cells lacking TRF2[B] and RAP1 (Fig. 1f, Supplementary Fig. 1e, f). Telomere bridge formation peaked at 24 and 48 h, disappeared at later time points and preceded the appearance of UTs (Fig. 1c, e, f). To determine whether the telomeric bridges observed at 48 hrs are composed of double-stranded or single-stranded telomeric DNA, we performed Immunofluorescence and fluorescence in situ hybridization (IF-FISH)[26] using the single-stranded DNA binding proteins Replication Protein 1 (RPA) and RAD51. We found that 100% of the telomeric bridges in *Rap1[−/−]* cells reconstituted with TRF2[ΔB] co-localized with RPA and RAD51, suggesting that telomere bridges consist of single-stranded telomeric DNA and that bridging between two telomeres involves proteins that participate in homology directed repair (HDR) (Fig. 1h, Supplementary Fig. 1g). Reconstitution of *Rap1[−/−]* MEFs with TRF2[ΔB] activated a robust ATR-dependent DDR, manifested as increased phospho-CHK1 staining (Supplementary Fig. 1h).

We hypothesized that formation of UTs is due to telomere bridging of multiple telomeres, resulting in their clustering into UTs[27]. The mobility of dysfunctional telomeres has been shown to be dependent on 53BP1's S-T/Q residues[28–31]. To test this hypothesis, we used *53Bp1[−/−]* MEFs where telomere mobility is eliminated. Compared to *53Bp1[+/+]* MEFs, *53Bp1[−/−]* MEFs reconstituted with TRF2[ΔB;L286R] do not form telomere bridges nor UTs (Fig. 1i, Supplementary Fig. 1i). Expression of WT 53BP1, 53BP1[15AQ] (15 conserved (S/T)Q phosphorylation sites replaced with AQ sites), 53BP1[OligoD1256A] and 53BP1[BRCTS1853A] mutants, but not the 53BP1[ΔNH3] or 53BP1[Δ28] mutants (all 28 conserved (S/T)Q phosphorylation sites replaced with AQ sites), restored the ability of *53Bp1[−/−]* MEFs reconstituted with TRF2[ΔB;L286R] to generate UTs (Fig. 1i, Supplementary Fig. 1j). Taken together, our results suggest that both RAP1 and the basic domain of TRF2 are required to repress 53BP1-dependent telomere bridging and the formation of UTs.

### Recruitment of homology directed repair factors to ultrabright telomeres

The localization of HDR factors RPA and RAD51 to telomeric filaments suggests that HDR is likely an early step in the generation of UTs. To test this hypothesis, we examined the localization of HDR proteins to telomeres in *Rap1[−/−]* MEFs reconstituted with TRF2[ΔB] or the telomerase-negative U2OS cell line[32] expressing TRF2[ΔB;L288R]. We found that in both cell lines, nearly 100% of UTs co-localized with HDR proteins p-RPA32 and RAD51 (Fig. 2a, b, Supplementary Fig. 2a, b). UTs decreased by ~3-fold in both RAD51 depleted MEFs and NBS1 null MEFs, but this decrease was not observed in the absence of the A-NHEJ factor Ligase 3 (Fig. 2b, c, Supplementary Fig. 2c). These results reveal the critical importance of HDR proteins in the formation of UTs. Another important HDR factor is SLX4, whose localization to telomeres is repressed by RAP1 and TRF2[B][18]. We therefore asked if SLX4 is required for the formation of UTs. U2OS cells devoid of SLX4 but reconstituted with TRF2[ΔB;L288R] completely lost the ability to form UTs, suggesting that SLX4 is critical for UT formation (Fig. 2d, Supplementary Fig. 2d).

DNA polymerase θ (POLθ) is a known inhibitor of RAD51-mediated HDR DNA repair[33,34]. To determine whether POLθ inhibits the

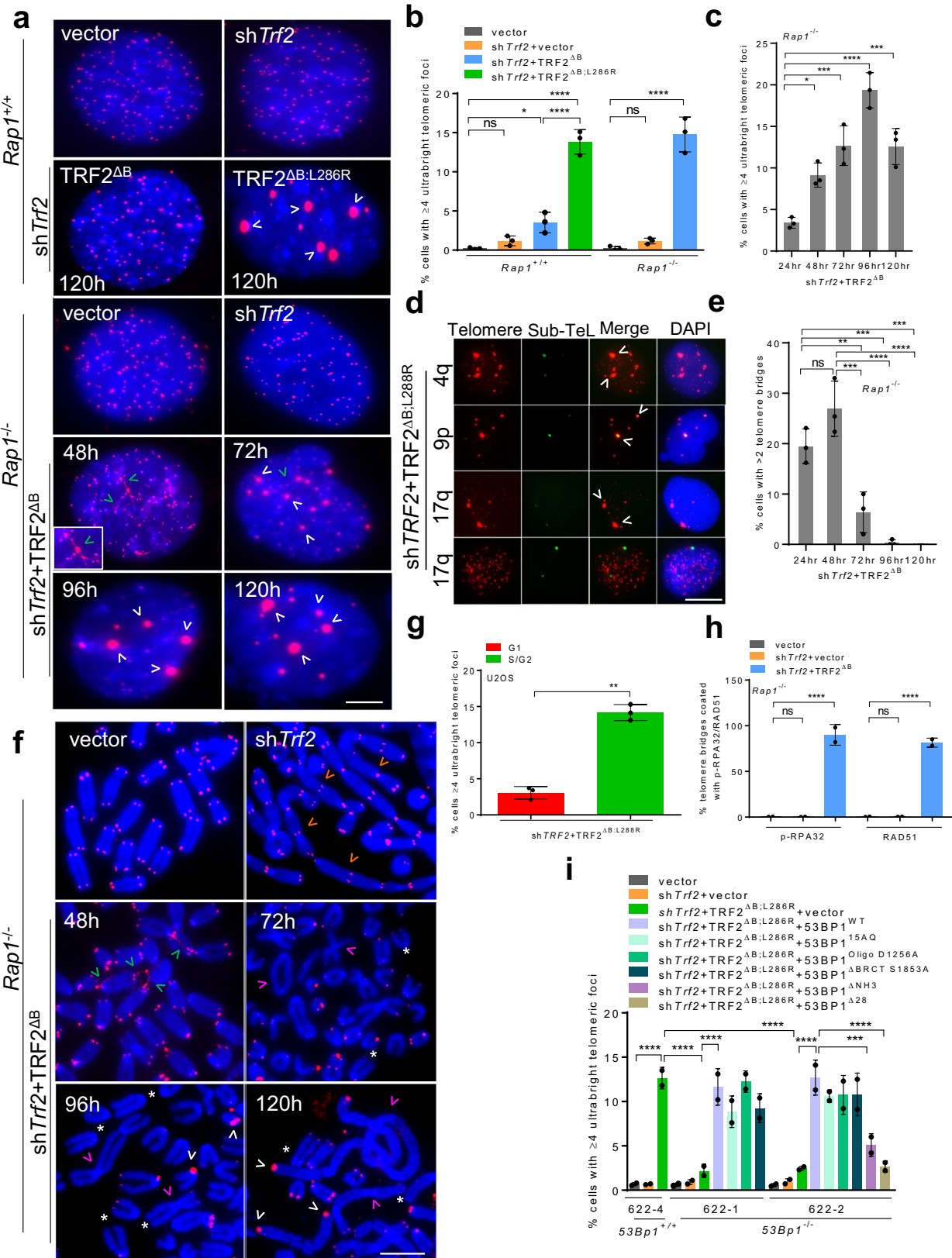

formation of UTs, we depleted *POLθ* in U2OS or in *Rap1⁻/⁻* MEFs and then reconstituted depleted cells with either TRF2^{ΔB;L288R} or TRF2^{ΔB}. Depletion of POLθ promoted an ~3-fold increase in the percent of cells bearing UTs in both U2OS and *Rap1⁻/⁻* MEFs (Fig. 2c, e, f, Supplementary Fig. 2c). Expression of WT POLθ in U2OS cells reduced the formation of UTs to basal levels (Fig. 2e, f). To ascertain which POLθ

domain is required to repress UT formation, we expressed a series of POLθ mutants in U2OS cells expressing TRF2^{ΔB;L288R}: mutants lacking the ATPase activity in its helicase-like domain (K121M), polymerase activity (D2330A/Y2331A), both helicase and polymerase activities (K121M;D2330A/Y2331A) and the S1977P mutant mimicking the chaos1 mutation[35]. Only the POLθ^{K121M} and the POLθ^{K121M;D2330A/Y2331A} double

**Fig. 1 | TRF2$^B$ and RAP1 cooperate to repress ultrabright telomere formation.** **a** PNA- FISH of interphase nuclei showing time-dependent increase of CCCTAA-positive telomere filaments (green arrowheads) and UT foci (white arrowheads) in *Rap1*$^{+/+}$ and *Rap1*$^{-/-}$ MEFs expressing the indicated DNAs. CCCTAA-positive filaments and telomeres detected with TelC-Cy3 (CCCTAA)$_3$ PNA telomere probe (red) and DAPI to stain nuclei (blue). Inset: magnified view of CCCTAA-positive filaments and UTs. Scale bars: 5 µm. **b** Quantification of UTs in *Rap1*$^{+/+}$ and *Rap1*$^{-/-}$ MEFs after 120 hrs. The mean of three independent experiments ± SD are shown, at least 200 nuclei analyzed per experiment. *$P = 0.0458$, ****$P < 0.0001$ by one-way ANOVA. ns: non-significant. **c** Quantification of UTs in *Rap1*$^{-/-}$ MEFs expressing TRF2$^{ΔB}$ at the indicated time points. The mean of three independent experiments ± SD are shown, at least 250 nuclei analyzed per experiment. *$P = 0.0240$, ***$P = 0.0010$, ****$P < 0.0001$ by one-way ANOVA. ns non-significant. **d** Representative images of UTs from two independent experiments shown to co-localized (white arrowhows) with indicated chromosomal specific sub-telomere probes (green) in U2OS cells expressing TRF2$^{ΔB;L288R}$. Scale bars: 5 µm. **e** Quantification of CCCTAA-positive filaments in *Rap1*$^{-/-}$ MEFs expressing TRF2$^{ΔB}$ at the indicated time points. The mean of three independent experiments ± SD are shown; at least 250 nuclei analyzed per experiment. **$P = 0.0060$, ***$P = 0.0003$, ****$P < 0.0001$ by one-way ANOVA. ns non-

significant. **f** Representative PNA-FISH on metaphase spreads of *Rap1*$^{-/-}$ MEFs expressing TRF2$^{ΔB}$ from three independent experiments showing maximum formation of CCCTAA-positive telomere filaments (red) at 48 hrs (green arrowheads), UTs (white arrowheads), signal free ends (*), chromosomes fused with telomeres (orange arrowheads), chromosomes fused without telomeres (pink arrowheads). A minimum of 35 metaphase for each sample were examined per experiment. Scale bars: 15 µm. **g** The Fucci assay showing UTs formed primarily during the S-phase in U2OS cells. The mean of three independent experiments ±SD are shown, at least 150 nuclei analyzed per experiment. **$P = 0.0015$ by two-tailed unpaired $t$ test. **h** Quantification of p-RPA32 and RAD51 localization on telomere bridges in *Rap1*$^{-/-}$ MEFs expressing indicated DNA constructs after 48 h. The mean of two independent experiments ± SD are shown, at least 200 nuclei analyzed per experiment. ****$P < 0.0001$ by one-way ANOVA. ns non-significant. **i** Quantification of UT frequencies in *53Bp1*$^{+/+}$ and *53Bp1*$^{-/-}$ MEFs reconstituted with WT 53BP1 and the indicated 53BP1 mutants in the presence of sh*Trf2* + TRF2$^{ΔB;L286R}$. The mean of two independent experiments ± SD are shown, at least 150 nuclei analyzed per experiment. ***$P = 0.0010$, ****$P < 0.0001$ by one-way ANOVA. Source data are provided as a Source Data file.

mutant were unable to repress the formation of UTs (Fig. 2f), suggesting that the helicase domain of POLθ is essential to repress UT formation. This result is consistent with the observation that helicase activity is required for inhibiting HDR-mediated DNA repair[33].

Using the TRF1-FokI system, previous studies have shown that telomeric HDR can be visualized as clustering of telomeres into larger telomeric foci[27,36]. To ascertain whether UTs in the absence of TRF2$^B$ and RAP1 resemble the TRF1-FokI induced telomere foci, we expressed WT TRF1-FokI and the nuclease dead TRF1-FokI$^{D450A}$ mutant in both U2OS cells and MEFs. Compared to cells expressing the TRF1-FokI$^{D450A}$ mutant, expression of WT TRF1-FokI resulted in an increase in average telomere foci size in U2OS cells (Fig. 2g, Supplementary Fig. 2e). However, TRF1-Fok1 induced telomere foci sizes were appreciably smaller than the UTs observed in the absence of TRF2$^B$ and RAP1 (Fig. 2g, Supplementary Fig. 2e). Telomeric filaments were also never observed in U2OS cells expressing TRF1-FokI (data not shown). In addition, TRF1-FokI induced telomere breaks trigger a 53BP1-independent canonical DSB response in cells utilizing the ALT pathway that activates primarily the ATM kinase[27,37]. In contrast, TRF2$^B$ and RAP1 loss activates the ATR-CHK1 kinase pathway (Supplementary Fig. 1h) and telomere clustering to form UTs in a 53BP1-dependent manner, suggesting distinctive mechanisms underlying telomere bridging and UT formation. Taken together, our data suggest that TRF2$^B$ and RAP1 cooperate to repress the localization of HDR factors to dysfunctional telomeres, preventing massive telomere clustering and the formation of UTs.

## TRF2$^B$ and RAP1 cooperate to repress ALT-associated proteins at telomeres

UT formation in the absence of TRF2$^B$ and RAP1 is reminiscent of the telomeric clustering observed in ALT cell lines[24,27,38]. We found that ALT cell lines U2OS, SAOS2 and CAL72 all exhibited significant increases in the number of UTs when reconstituted with TRF2$^{ΔB; L288R}$ (Fig. 3a). A hallmark of ALT-like telomeric clustering is the loss of ATRX and formation of ALT-associated PML bodies (APBs) on telomeres. We postulated that TRF2$^B$ and RAP1 cooperate to repress the localization of ALT-associated factors to telomeres to prevent HDR mediated telomere clustering. In support of this notion, we found that PML localized to nearly 100% of UTs in U2OS cells reconstituted with TRF2$^{ΔB; L288R}$ (Fig. 3b, Supplementary Fig. 3a). In addition, sh*Atrx* mediated depletion of endogenous ATRX further increased the formation of UTs by an additional 2-fold in *Rap1*$^{-/-}$ cells expressing TRF2$^{ΔB}$ (Fig. 3c−e). Given these findings, we reasoned that ATRX-null ALT cell lines should display increased UTs as compared to ATRX-positive non-ALT cell lines. We found that ATRX-null cell lines U2OS, SAOS2 and CAL72 exhibited significantly higher numbers of UTs compared to the ATRX-positive

MG63 cells, while ATRX-negative U2OS cells reconstituted with GFP-ATRX displayed a ~3-fold decrease in the number of UTs observed (Fig. 3a, Supplementary Fig. 3b, c).

Break-induced replication (BIR) has been shown to be important to mediate TRF1-FokI mediated ALT telomere clustering[27,36,39]. BIR can arise from either RAD51-dependent or RAD52-dependent pathways[27,36,39]. We have already shown that formation of UTs requires RAD51 (Fig. 2b). shRNA mediated knockdown of RAD52 in U2OS cells expressing TRF2$^{ΔB;L288R}$ did not impact the formation of UTs, suggesting that these structures arose by a RAD51-dependent, RAD52-independent repair pathway (Fig. 3f, Supplementary Fig. 3d). To address whether other BIR components are involved in the generation of UTs, we performed knockdown of POLD1, the catalytic subunit of POLD, using three independent shRNAs. Surprisingly, depletion of POLD1 reduced the number of p-RPA32 (S33) positive TIFs and the number of UTs in *Rap1*$^{-/-}$ MEFs expressing TRF2$^{ΔB}$, suggesting a role for POLD1 in the formation of UTs (Supplementary Fig. 3e−h). These finding are consistent with the role for POLD subunits in HDR[40,41]. Since UTs were observed in both ALT-positive and ALT-negative cells, we examined for ALT phenotypes in telomerase positive *Rap1*$^{-/-}$ MEFs expressing TRF2$^{ΔB}$. Using the C-circle assay[42], we found a ~3-fold increase in the amount of C-circles in *Rap1*$^{-/-}$ MEFs expressing TRF2$^{ΔB}$ as compared to vector control (Supplementary Fig. 3i, j). However, the amount of C-circles observed in *Rap1*$^{-/-}$ MEFs is ~2.5-fold less compared to those observed in a genuine ALT cell line (U2OS) (Supplementary Fig. 3j). In addition, both ALT-positive and ALT-negative cells are known to promote mitotic DNA synthesis (MiDAS) at telomeres[38,43]. We therefore asked whether UT formation due to loss of TRF2$^B$ and RAP1 induces MiDAS. We observed increased EdU foci at single or both chromatids in *Rap1*$^{-/-}$ MEFs expressing TRF2$^{ΔB}$, suggesting that TRF2$^B$ and RAP1 normally repress replication stress-induced MiDAS (Supplementary Fig. 3k, l). Together, our data reveal that TRF2$^B$ and RAP1 repress ALT-specific phenotypes and the localization of ALT-specific proteins to telomeres.

## Ultrabright telomeres are sources of replication stress

The observed increase in recruitment of RPA, RAD51 and other HDR factors to UTs in both ALT-negative and ALT-positive cells (Fig. 2, Supplementary Fig. 4a, b) suggests that UT formation might be an adaptive response to stalled replication forks at telomeres[27]. In support to this notion, we found that UTs colocalize to EdU containing p-RPA32 (S33) foci in *Rap1*$^{-/-}$ MEFs expressing TRF2$^{ΔB}$, suggesting that they are the product of replication intermediates (Supplementary Fig. 4c, d). To test this hypothesis, we treated WT and *Rap1*$^{-/-}$ MEFs reconstituted with either TRF2$^{ΔB;L286R}$ or TRF2$^{ΔB}$, respectively, with aphidicolin (APH) to increase replication stress or vehicle control. APH treatment

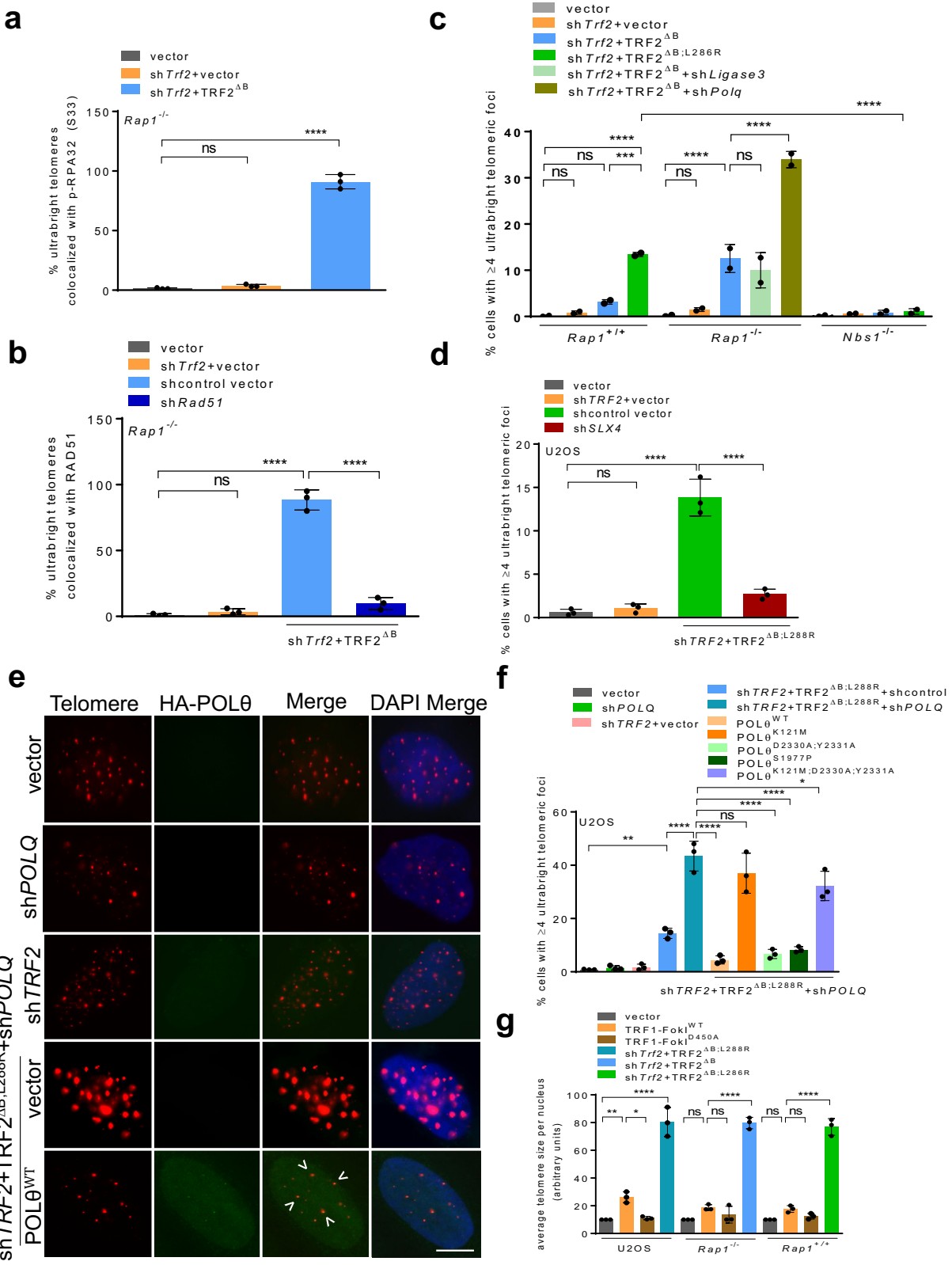

increased the frequency of telomere bridging and UTs by ~2-fold (Fig. 4a, b, Supplementary Fig. 4e). We next examined the status of SMARCAL1 (SWI/SNF-related, matrix-associated, actin-dependent regulator of chromatin, subfamily A-like 1), an ATP-dependent annealing helicase that localizes to stalled replication forks to promote replication restart[44–46]. SMARCAL1 also localizes to telomeres

undergoing replication stress[45,46]. Since APH treatment significantly increased telomere bridging and UT formation, we asked whether SMARCAL1 localized to UTs. While cells lacking TRF2 displayed only background levels of SMARCAL1-positive TIFs, ~15% of U2OS cells expressing TRF2$^{\Delta B;L288R}$ displayed >5 SMARCAL1-positive TIFs (Fig. 4c, d). Importantly, nearly 100% of UTs co-localized with

**Fig. 2 | Recruitment of HDR factors promote the formation of ultrabright telomeres. a** Quantification of percent UTs co-localized with p-RPA32 (S33) in *Rap1*⁻/⁻ MEFs expressing TRF2^ΔB. Data represents the mean of three independent experiments ± SD from a minimum 200 nuclei analyzed per experiment. ****$P < 0.0001$ by one-way ANOVA. ns non-significant. **b** Quantification of percent UTs co-localized with RAD51 in *Rap1*⁻/⁻ MEFs expressing TRF2^ΔB. Data represents the mean of three independent experiments. At least 200 nuclei were examined per experiment. ****$P < 0.0001$ by one-way ANOVA. ns non-significant. **c** Quantification of UT frequencies in the absence of NBS1 and Ligase 3 that promote A-NHEJ. Data represents the mean values from two independent experiments ± SD from a minimum 200 nuclei analyzed per experiment. ***$P = 0.0005$, ****$P < 0.0001$ by one-way ANOVA. ns non-significant. **d** Quantification of the percentage of U2OS cells possessing UTs with or without treatment with *SLX4* shRNA. Data represents the mean of three

independent experiments ± SD from a minimum 200 nuclei analyzed per experiment. ****$P < 0.0001$ by one-way ANOVA. ns non-significant. **e** IF-FISH for HA-POLθ (green) co-localizing with telomeres (red) in U2OS cells expressing TRF2^ΔB,L288R. Scale bars: 5 μm. **f** Quantification of UTs in U2OS cells expressing the indicated DNA constructs. Data represents the mean of three independent experiments ±SD from a minimum 150 nuclei analyzed per experiment. *$P = 0.0295$, **$P = 0.0051$, ****$P < 0.0001$ by one-way ANOVA. ns non-significant. **g** Quantification showing average telomere size per nucleus in U2OS, *Rap1*⁻/⁻ and *Rap1*⁺/⁺ MEFs expressing indicated DNA constructs. Data represents the mean of three independent experiments ±SD from a minimum 100 nuclei analyzed per experiment. *$P = 0.01$, **$P = 0.0064$, ****$P < 0.0001$ by one-way ANOVA. ns non-significant. Source data are provided as a Source Data file.

SMARCAL1 (Fig. 4c). These results suggest that UTs are regions of stalled telomere replication.

Telomere restriction fragment (TRF) Southern blot under native conditions reveal that *Rap1*⁻/⁻ MEFs reconstituted with TRF2^ΔB display ss telomeric DNA of very heterogeneous lengths that is resistant to exonuclease I (Exo I) digestion, suggesting that these structures are not composed of the usual ss telomeric overhangs (Fig. 4e, f, Supplementary Fig. 4f). Under denaturing conditions, TRF Southern blots also show telomeric DNA trapped in the loading well. These results suggest that in the absence of TRF2^B and RAP1, telomeric DNA adopt complex secondary structures that hinder gel migration. Telomeres can adopt a telomeric (T)-complex composed of branched telomere DNA structures and recombination intermediates, leading to the formation of slower migrating DNA in two-dimensional gel electrophoresis[47]. These highly branched telomeric structures contain large numbers of internal ss portions sensitive to T7 endonuclease I but not to Exo I. A potential source of T-complex is multiple telomeres undergoing HDR with each other[47]. Our metaphase analysis suggests that telomere bridging and the formation of UTs involves multiple chromosomes and occurs through an inter-telomeric recombination mechanism (Fig. 4a). TRF Southern analysis revealed that telomeric DNA trapped in the loading wells is sensitive to T7 endonuclease I but not to Exo I (Fig. 4e, f, Supplementary Fig. 4f). To ascertain whether telomere bridging and UT formation is a result of inter-telomeric recombination leading to branched DNA structures and unresolved recombination intermediates, we performed telomere two-dimensional gel electrophoresis[48] on *Rap1*⁻/⁻ MEFs treated with vector, sh*Trf2* or sh*Trf2* + TRF2^ΔB. Expression of TRF2^ΔB in *Rap1*⁻/⁻ MEFs led to the formation of a slow migrating T-complex that is sensitive to T7 endonuclease I, and its level further increased by 2-fold after APH treatment (Fig. 4g, h). T-complexes increased in *Rap1*⁻/⁻ MEFs expressing TRF2^ΔB in a time-dependent manner, supporting the notion that increased UT size is due to increased telomere recombination (Supplementary Fig. 4g, h). In addition, we also found an increase in the number of telomere circles (TCs) formed at 120 hrs (Supplementary Fig. 4g). An in vitro TC assay also revealed that increased telomere recombination in *Rap1*⁻/⁻ MEFs expressing TRF2^ΔB promoted the accumulation of TCs in a time-dependent manner (Supplementary Fig. 4i). These data suggest that TRF2^B and RAP1 repress the formation of unresolved telomere replication intermediates, thereby preventing the initiation of t-loop HDR and rapid telomere loss due to excision of telomeres as TCs[18].

We have recently shown that the protein DONSON forms a complex with the replisome proteins Claspin and Proliferating Cell Nuclear Antigen (PCNA), and that this complex interacts with TRF2 to promote telomere replication and the generation of the 3′ ss overhang after replication[49]. We hypothesized that replication fork stalling could be a critical early step in the formation of UT foci and depletion of any of CPD (Claspin, PCNA, DONSON) components would lead to increased frequency of replication fork stalling to enhance UT formation. To test this hypothesis, CPD components were individually depleted in *Rap1*⁻/⁻

MEFs expressing sh*Trf2* + TRF2^ΔB. Depletion of individual CPD components, combined with the loss of both TRF2^B and RAP1, increased UTs by ~2-fold (Supplementary Fig. 4j, k). UT formation requires EXO1, consistent with the need for EXO1 to generate ss overhangs important for HDR (Supplementary Fig. 4j, k). Together, our data suggest that increased recruitment of HDR factors to stalled telomere replication forks is a prerequisite for UT formation.

## TRF2^B and RAP1 cooperate to repress TERRA and R-loop formation in ultrabright telomeres

Accumulation of R-loops consisting of DNA-RNA hybrids on telomeres promotes HDR and represents a source of telomere replication stress[50,51]. To determine whether TRF2^B and RAP1 cooperate to modulate R-loop formation at telomeres, we examined for the presence of R-loops at UTs using the S9.6 antibody that recognizes DNA-RNA hybrids[52]. Compared to vector or *shTrf2* treated controls, R-loops were significantly enriched on UTs in *Rap1*⁻/⁻ MEFs reconstituted with TRF2^ΔB, with ~80% of all UTs staining positive with S9.6 (Fig. 5a, b). TERRA association with telomeres has been shown to stimulate R-loop formation[51,53]. To assess whether R-loops at UTs are formed by TERRA, we performed TERRA FISH in *Rap1*⁺/⁺ and *Rap1*⁻/⁻ MEFs expressing TRF2^ΔB. *Rap1*⁻/⁻ MEFs expressing TRF2^ΔB displayed a two-fold increase in TERRA co-localization with UTs. Importantly, ~80% of UTs show co-localization with TERRA (Fig. 5c, d). RNaseH treatment dramatically reduced TERRA foci formation, demonstrating that TERRA foci are composed of DNA-RNA hybrids (Supplementary Fig. 5a, b). Doxycycline-induced expression of WT RNaseH1 but not the RNaseH1^D210N catalytically dead mutant significantly reduced UT formation (Fig. 5e)[54,55]. Expression of RNaseH1^WT but not RNaseH1^D210N also abolished S9.6 foci, revealing the specificity of S9.6 antibody for DNA-RNA hybrids (Supplementary Fig. 5c, d). Accumulation of R-loops also correlated with high TERRA levels at UTs (Supplementary Fig. 5e, f). Finally, overexpression of WT RNaseH1 but not the RNaseH1^D210N reduced the co-localization of SMARCL1 with UTs and also reduced the formation of the T-complex, further strengthening the notion that UTs contain R-loops (Supplementary Fig. 5g–j). Together, these results suggest that TRF2^B and RAP1 play important roles in counteracting TERRA-dependent R-loop formation at telomeres.

To determine whether TRF2 specifically interacts with other R-loop resolving proteins at telomeres, we performed immunoprecipitation-mass spectrometry (IP-MS) using Flag-TRF2 and Flag-vector as baits. IP-MS data revealed that a number of R-loop associated proteins co-precipitated with Flag-TRF2 (Fig. 5f). DDX21, a DEAD-box RNA helicase known to unwind R-loops[56], was significantly enriched in TRF2-IP samples (Fig. 5f). Co-IPs of WT TRF2 and various TRF2 deletion mutants with DDX21 shows that removal of TRF2^B severely disrupted interaction with DDX21, suggesting that TRF2^B mediates the interaction with DDX21 (Supplementary Fig. 5k). In a converse experiment, we found that the DDX21^584−783 C-terminal domain is required to interact with WT TRF2 (Fig. 5g). We hypothesize that recruitment of DDX21 to telomeres might be required to

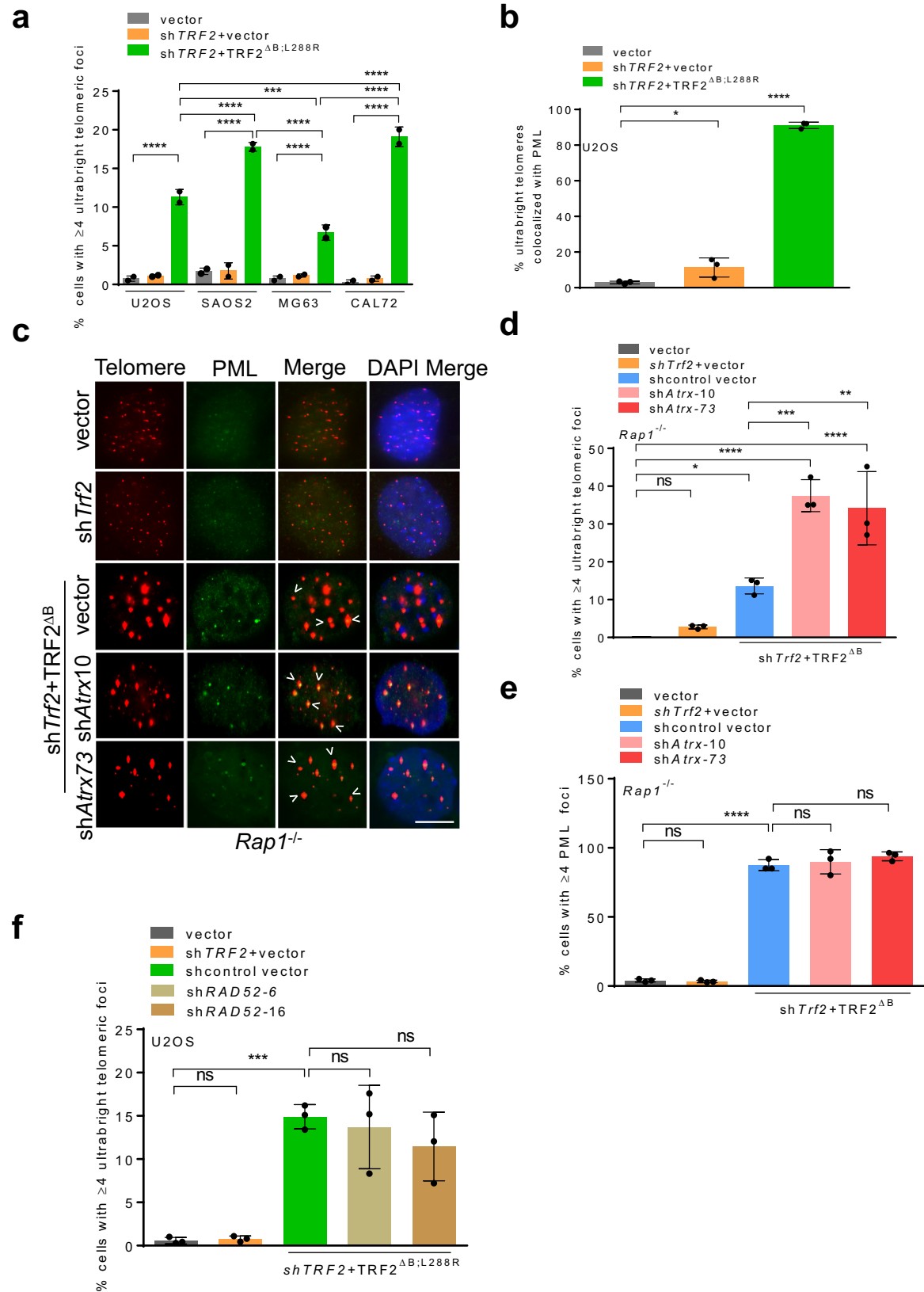

suppress R-loop formation at telomeres. To test this hypothesis, we overexpressed WT Flag-DDX21 or DDX21 domain specific mutants in U2OS cells expressing TRF2^{ΔB; L288R}. Only WT DDX21 was able to completely repress UT formation (Fig. 5h). IP-MS revealed a second TRF2-interacting protein, ADAR1p110, a protein known to edit the A-C mismatches within DNA-RNA telomere hybrids to facilitate resolution of

telomeric R-loops (Fig. 5f)[57]. We found that the TRF2^B also mediates the interaction between TRF2 and ADAR1p110 (Supplementary Fig. 5l). Compared to TRF2^{ΔB}, WT TRF2 interacts three times stronger with WT ADAR1p110 (Supplementary Fig. 5l). Interaction between WT TRF2 and WT ADAR1p110 is independent of ADARp110's catalytic activity, since the catalytically dead ADAR1p110^{E912A} mutant interacted with WT TRF2

**Fig. 3 | TRF2$^B$ and RAP1 repress ALT associated proteins at telomeres.**
**a** Quantification of UT frequencies in ATRX-positive and ATRX-null ALT cells expressing indicated DNA constructs. Data represents the mean of two independent experiments ±SD from a minimum 200 nuclei analyzed per experiment. ***$P = 0.0009$, ****$P < 0.0001$ by one-way ANOVA. **b** Quantification of percent UTs co-localized to PML bodies in U2OS expressing sh*TRF2* + TRF2$^{ΔB;L288R}$. Data represents the mean of three independent experiments ±SD from a minimum 150 nuclei analyzed per experiment. *$P = 0.0460$, ****$P < 0.0001$ by one-way ANOVA.
**c** Representative IF-FISH images showing the loss of ATRX promotes the formation of UTs colocalized with PML (green) in *Rap1*$^{-/-}$ MEFs expressing TRF2$^{ΔB}$. Scale bars: 5 μm. **d** Quantification of cells containing ≥4 UTs with or without *Atrx* shRNAs in (**c**).

Data represents the mean of three independent experiments ± SD from a minimum 250 nuclei analyzed per experiment. *$P = 0.0407$, **$P = 0.0028$, ***$P = 0.0009$, ****$P < 0.0001$ by one-way ANOVA. ns: non-significant. **e** Quantification of percent UTs co-localized to PML bodies in (**c**). Data represents the mean of three independent experiments ± SD from a minimum 250 nuclei analyzed per experiment. ****$P < 0.0001$ by one-way ANOVA. ns non-significant. **f** Quantification of cells containing ≥4 UTs with or without *RAD52* shRNAs. The mean of three independent experiments ± SD shown from a minimum 200 nuclei analyzed per experiment. ***$=0.0008$ by one-way ANOVA. ns non-significant. Source data are provided as a Source Data file.

---

in a manner similar to WT ADAR1p110 (Supplementary Fig. 5m). We also found that overexpression of WT ADAR1p110, but not the catalytically inactive ADAR1p110$^{E912A}$ mutant, repressed the formation of UTs (Fig. 5i). Taken together, these results suggest that TRF2 recruits DDX21 and ADAR1p110 to resolve DNA-RNA hybrids at telomeres, preventing the formation of telomeric R-loops and UTs.

### The RAP1$^{BRCT}$ domain interacts with KU70-KU80 to repress ultrabright telomere formation

We have shown previously that both RAP1$^{Myb}$ and RAP1$^{BRCT}$ domains are required to repress the formation of telomere-free fusions and signal-free ends in cells expressing TRF2$^{ΔB;L286R}$[18]. To elucidate which RAP1 domain(s) are able to cooperate with TRF2$^B$ to repress telomere bridging and UT foci, we reconstituted *Rap1*$^{-/-}$ MEFs with various RAP1 proteins tethered to TRF2$^{ΔB}$, including WT RAP1 and RAP1 mutants lacking either the BRCT, Myb or the C-terminal (RCT) domains[18]. Overexpression of WT RAP1, RAP1$^{ΔMyb}$, RAP1$^{ΔLinker}$ and RAP1$^{ΔRCT}$ all resulted in dramatic reductions in the number of UTs observed (Fig. 6a, b). In contrast, expression of RAP1$^{ΔBRCT}$ failed to repress filamentous telomere bridging and UT formation. Examination of metaphase spreads revealed that the RAP1$^{BRCT}$ domain is also required to repress the generation of telomere-free end-to-end chromosome fusions and signal-free chromosome ends (Fig. 6c, Supplementary Fig. 6a). Together, these results suggest that the RAP1$^{BRCT}$ domain cooperates with TRF2$^B$ to repress telomere bridging and UT formation. While the function of the RAP1$^{BRCT}$ domain is largely unknown, it is structurally very similar to the BRCT modules of other damage repair proteins that mediate protein-protein interactions[58]. This result suggests that RAP1$^{BRCT}$ might participate in physical interactions with proteins involved in DNA damage and repair.

To search for potential RAP1$^{BRCT}$ interacting partners which might repress the formation of UTs, we first performed GST pull-down assays from 293 F cell lysates by using GST-RAP1$^{WT}$ and GST-RAP1$^{ΔBRCT}$ as baits, and then used mass spectrometry to identify the proteins in the pulldown samples. We found that KU70, KU80, and PRKDC specifically associated with GST-RAP1$^{WT}$, but not with GST-RAP1$^{ΔBRCT}$, suggesting a potential physical interaction between RAP1$^{BRCT}$ and KU70-KU80 (Supplementary Fig. 6b). To test this hypothesis, purified GST-RAP1$^{WT}$ and GST-RAP1$^{ΔBRCT}$ were subjected to a pull-down assay with 293 T cell lysates with over-expressed HA-KU70 and Myc-KU80. HA-KU70 and Myc-KU80 were both pulled down by GST-RAP1$^{WT}$ but not GST-RAP1$^{ΔBRCT}$, indicating a possible direct RAP1-KU70/KU80 interaction mediated by the BRCT domain (Fig. 6d). Since the BRCT domain is a canonical phosphorylation reader, to ascertain whether phosphorylation of the KU70/80 is required for binding to RAP1$^{BRCT}$ domain, lysates were treated with lambda protein phosphatase. Both HA-KU70 and Myc-KU80 signals pulled down by GST-RAP1$^{WT}$ were significantly reduced in the presence of Lambda-PPase (Fig. 6d). Thus, RAP1$^{BRCT}$ interaction with KU70-KU80 is dependent on the phosphorylation status of KU70/KU80.

Both RAP1 and the KU70/KU80 complex have been shown to effectively suppress HDR at chromosome ends[11,12,18]. We hypothesized that the physical interaction between KU70/KU80 and RAP1$^{BRCT}$

enables this complex to repress HDR-mediated telomere recombination and the generation of UTs, and that disruption of KU70/KU80 RAP1 interaction would further exacerbate telomere bridging and UT formation. To test this hypothesis, we reconstituted TRF2$^{ΔB;L286R}$ in cells lacking *Ku70/Ku80*. Reconstitution of TRF2$^{ΔB;L286R}$ in *Ku70*$^{-/-}$/*Ku80*$^{-/-}$ MEFs resulted in dramatically increased levels of telomere bridging and UT formation (Fig. 6e, f). Taken together, our data suggest that physical interaction between KU70/KU80 and the RAP1$^{BRCT}$ domain is important for repressing telomere bridging and UT formation.

### Depletion of lamin A promotes the formation of ultrabright telomeres

An unexpected phenotype observed in *Ku70*$^{-/-}$ MEFs expressing TRF2$^{ΔB;L286R}$ is defects in the nuclear envelope (NE), manifested as irregular invaginations and blebbing after staining with an anti-lamin A antibody. In addition, discontinuous lamin A staining characterized by one or more large holes in the lamina was observed (Supplementary Fig. 6c). These holes are not repaired and result in NE rupture (Supplementary Fig. 6d). In metazoans, a family of A-type and B-type lamins form a meshwork of intermediate filaments underlying the inner membrane of the NE to maintain nuclear shape, mechano-signaling and spatial organization within the nucleus[59,60]. In addition to a structural role, nuclear lamins also play major roles in genome organization. Lamin A participate in NHEJ and HDR-mediated DSB repair by suppressing the mobility of broken DNA ends[61–65]. Altered lamin homeostasis leads to telomere shortening, telomere mis-localizations and defects in DNA damage response suggesting a role of lamin A in telomere maintenance[61,66–68]. RAP1 interacts with SUN1, a protein in the NE that interacts with lamin A[3]. We hypothesized that this interaction might be required to restrict the mobility of damaged telomeres, preventing homology search and UT clustering observed in *Rap1*$^{-/-}$ MEFs reconstituted with TRF2$^{ΔB}$. We found that lamin A localizes to the nuclear periphery in cells expressing either vector or sh*Trf2*. However, in cells bearing UTs, we detected irregular lamin A localization, manifested as numerous NE infoldings, blebbing and holes characteristic of NE rupture (Fig. 7a, b)[69–72]. NE rupture sites promote the localization of cytosolic GMP-AMP synthase (cGAS), a cytosolic DNA sensor[73]. GFP-cGAS localized to micronuclei upon NE rupture after 1 Gy irradiation used as positive control (Supplementary Fig. 6e). Consistent with this observation, we found increased localization of cGAS at NE rupture sites, with ~25% of GFP-cGAS co-localizing with telomeres (Supplementary Fig. 6f, g). Overexpression of WT lamin A reduced the number of UTs by two-fold and restored normal lamin A nuclear architecture (Fig. 7a–c). In contrast, overexpression of a truncated form of lamin A known as progerin, found in the premature aging syndrome Hutchinson-Gilford progeria[69,74], did not decrease the frequency of UT formation in *Rap1*$^{-/-}$ MEFs expressing TRF2$^{ΔB}$ (Fig. 7a–c). Very few telomere bridging and UTs were detected in *Lmna*$^{-/-}$ MEFs; however, these aberrant structures increased when *Lmna*$^{-/-}$ MEFs were reconstituted with TRF2$^{ΔB;L286R}$ (Fig. 7d, Supplementary Fig. 7a). Overexpression of WT lamin A but not Progerin in these cells abolished the formation of both UTs and telomere bridging, suggesting an important

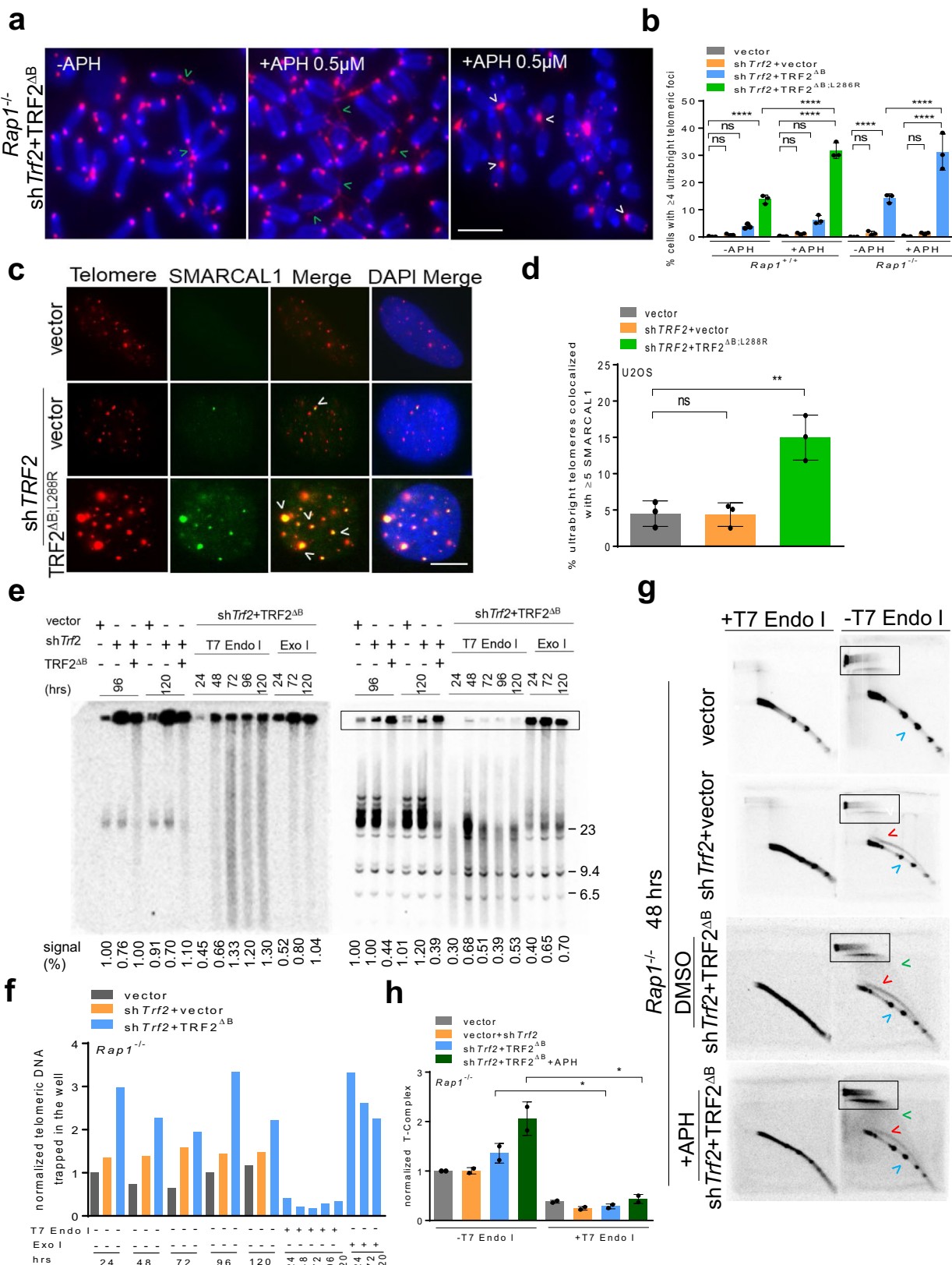

role for lamin A in repressing telomere HDR (Fig. 7d, Supplementary Fig. 7a).

Lamin A is subjected to multiple post-translational modifications throughout the cell cycle by numerous kinases and phosphatases[75–77]. Cell cycle dependent lamin A phosphorylation-dephosphorylation cycles are known to modulate its assembly/disassembly at the nuclear

lamina[76,78,79]. In addition to cGAS, we also found p-RPA32 (S33) localized to NE rupture sites in $Rap1^{-/-}$ MEFs expressing TRF2$^{\Delta B}$ (Supplementary Fig. 6h, i), suggesting involvement of the ATR-CHK1 DNA damage pathway. Since ATR is known to maintain NE integrity from mechanical or replication stress[80–82], we hypothesize that ATR-CHK1 activation in the absence of TRF2$^B$ and RAP1 might modulate NE and

**Fig. 4 | Replication defect is an early step in ultrabright telomere formation.**
**a** PNA-FISH on metaphase spreads showing CCCTAA-positive filaments (green arrowheads) and UTs (white arrowheads) in *Rap1*$^{-/-}$ MEFs expressing TRF2$^{ΔB}$ with +/− 0.5 μM aphidicolin (APH) treatment. Scale bars: 20 μm. A minimum of 30 metaphases for each sample were examined per experiment. **b** Quantification of UT frequencies in the interphase nuclei of *Rap1*$^{+/+}$ and *Rap1*$^{-/-}$ MEFs expressing the indicated DNA constructs in (**a**). Mean of three independent experiments ± SD are shown from a minimum 250 nuclei analyzed per experiment. ****$P$ < 0.0001 by one-way ANOVA. ns non-significant. **c** Representative IF-FISH images showing UTs co-localized to SMARCAL1 (green) in U2OS cells expressing sh*TRF2* + TRF2$^{ΔB;L288R}$. Scale bars: 5 μm. **d** Quantification of percent UTs colocalized with SMARCAL1 in (**c**). Mean of three independent experiments ± SD from a minimum 300 nuclei analyzed per experiment. **$P$ = 0.0032 by one-way ANOVA. ns non-significant. **e** In-gel single-strand G overhang (native, left panel) and total telomere (denatured, right panel) analysis in *Rap1*$^{-/-}$ MEFs expressing the indicated cDNA constructs at the indicated time point with ±T7 endonuclease I (T7 Endo I) and Exo I treatment. Single-strand G overhang and total telomere signals in vector was set at 100% after normalizing each lane with ethidium bromide staining that served as an internal loading control. The numbers below the gel represents single-strand G overhang and total telomere signals. Molecular weights are displayed on the right. **f** Quantification of normalized telomeric DNA trapped in the well (outlined by the box) at indicated time point in (**e**) from one representative experiment (see also Supplementary Fig. 4f). Signal intensities in vector was set at 100% after normalizing with sub telomeres signals that served as an internal loading control. **g** T-complex and T-circle analysis using two-dimensional (2D) gel electrophoresis from genomic DNA isolated from *Rap1*$^{-/-}$ MEFs expressing the indicated DNA constructs in ±0.5 μM APH treated with 40 units of T7 endonuclease I. T-complex (outlined by the box), t-circle (green arrowheads), ss G (red arrowheads), ds TRF (blue arrowheads). **h** Quantification of T-complex in (**g**). T-complex signal intensities in vector was set at 100% after normalizing with sub telomeres signals that served as an internal loading control. The graph represents the mean ± SD from two independent experiments. *$P$ = 0.0175 by two tailed unpaired *t* test. Source data are provided as a Source Data file.

lamin A function. Since lamin A$^{S395}$ is the predicted ATR phosphorylation site (using phospho ELM, a database of S/T/Y phosphorylation sites), we postulated that lamin A$^{S395}$ phosphorylation by ATR promotes lamin A disassembly, leading to increased inter-telomere homology search and telomere pairing during HDR. To test this hypothesis, we expressed the lamin A phosphomimetic mutant S395D or the phosphorylation deficient S395A mutants in U2OS cells reconstituted with vector or TRF2$^{ΔB;L288R}$. We then treated these cells with 4 different ATR inhibitors: AZD6738, AZ20, VE-822 and BAY1895344. Expression of either WT GFP-lamin A or the phosphorylation defective lamin A$^{S395A}$ resulted in their localization predominantly to the nuclear lamina and a ~3-fold reduction in the number of UTs observed (Fig. 7e, f). In contrast, in cells expressing the phosphomimetic lamin A$^{S395D}$, we detected intense punctate nucleoplasm staining without reduction in the number of UTs observed. Compared to vehicle treated cells, ATR inhibition greatly reduced lamin A$^{S395D}$-positive punctate nucleoplasm staining, accompanied by a significant reduction in the number of UTs (Fig. 7g, Supplementary Fig. 7b). In addition, telomere bridging and HDR-induced telomere deletion were all abolished by ATR inhibitors in *Rap1*$^{-/-}$ MEFs expressing TRF2$^{ΔB}$ (Supplementary Fig. 7c–e). While these results do not definitively prove that the ATR/CHK1 kinases directly phosphorylates lamin A, they suggest that the ATR-CHK1 pathway plays a role in UT formation.

CDK1-mediated phosphorylation of lamin A$^{S392}$ has been shown to promote lamin A disassembly[78]. To ascertain whether CDK1-dependent lamin A$^{S392}$ phosphorylation also promotes the formation of UTs, we expressed either the GFP-lamin A$^{S392D}$ CDK1 phosphomimetic or the CDK1-dependent phosphorylation deficient lamin A$^{S392A}$ mutant in U2OS cells reconstituted with vector or TRF2$^{ΔB;L288R}$. GFP-lamin A$^{S392A}$ localized predominantly to the nuclear lamina and resulted in a 4-fold reduction of UTs (Fig. 7e, f). In contrast, cells expressing GFP-lamin A$^{S392D}$ displayed intense punctate nucleoplasm without significant reduction in the number of UTs (Fig. 7e, f). To further confirm that CDK1-dependent phosphorylation of lamin A$^{S392}$ promotes the formation of UTs, we treated *Rap1*$^{-/-}$ MEFs expressing TRF2$^{ΔB}$ with CDK1 specific inhibitor Ro-3306. Inhibition of CDK1 kinase activity with Ro-3306 or Roscovitine completely abolished telomere bridging, UTs and telomere deletions (Fig. 7h, i, Supplementary Fig. 7f, g). We next asked whether the ATR-CHK1 and CDK1-mediated phosphorylation of lamin A is epistatic in the formation of UTs. To address this question, we expressed either the GFP-WT lamin, GFP-lamin A$^{S392D;S395D}$ double phosphomimetic mutant or the double phosphorylation deficient GFP-lamin A$^{S392A;S395A}$ mutants in U2OS cells reconstituted with vector or TRF2$^{ΔB;L288R}$. Similar to single mutants, the GFP-lamin A$^{S392A;S395A}$ molecule localized predominantly to the nuclear lamina, leading to an 8-fold reduction of UTs (Fig. 7e, f). However, overexpression of GFP-lamin A$^{S392D;S395D}$ resulted in intense punctate nucleoplasm formation, leading to a 2-fold increase in the formation of UTs over GFP-WT lamin A (Fig. 7e, f). These data suggest that lamin A and an intact NE is required to repress telomere bridging and UT formation in the absence of RAP1 and TRF2$^{B}$.

## Discussion

The mammalian shelterin complex is vital to protect telomeric ends from initiating aberrant DNA damage responses. We have shown earlier that catastrophic telomere loss and telomere-free fusions in the absence of RAP1 and TRF2$^{B}$ is due to aberrant telomere HDR[18]. The underlying mechanisms as to how telomere HDR leads to catastrophic telomere loss and telomere-free fusions were unclear. The data presented here reveal that mammalian RAP1 and TRF2$^{B}$ are essential to repress telomere HDR. Expression of TRF2$^{ΔB}$ in cells devoid of RAP1 results in massive telomere R-loop-induced telomere clustering and the formation of UTs. TRF2's interactions with ADAR1p110 and DDX21, two proteins that process RNA-DNA hybrids, are essential to repress UT formation. In addition, analyses of proteins that interact with RAP1 reveal a role for TRF2-RAP1 in repressing telomere HDR through interactions with NE proteins. Expressing lamin A serine 392 and 395 phosphomimetic mutants in *Rap1*$^{-/-}$ MEFs reconstituted with TRF2$^{ΔB}$ resulted in NE rupture and dramatically increased telomere bridging and UT formation. These data suggest that in the absence of TRF2$^{B}$ and RAP1, CDK1-dependent phosphorylation of lamin A perturbs the nuclear lamina, resulting in aberrant HDR-mediated UT formation (Fig. 8). Our results highlight the importance of the basic domain of TRF2, RAP1 and the nuclear envelope scaffold in repressing aberrant telomere-telomere recombination to maintain telomere homeostasis.

### TRF2$^{B}$ and RAP1 repress telomere secondary structures
Telomeres are prone to forming DNA/RNA secondary structures including T-loops, D-loops, G-quadruplexes, and R-loops[83,84]. T-loops resemble branched DNA structures, and TRF2$^{B}$ promotes the formation of T-loop to repress PARP1 activation and branch migration, thus preventing the formation of a double Holliday Junction (HJ) to prevent the deleterious cleavage of T-loops[9,13–18]. We found that telomeric DNA in *Rap1*$^{-/-}$ cells expressing TRF2$^{ΔB}$ forms branched telomere DNA structures (T-complexes) and recombination intermediates sensitive to the T7 endonuclease I but not ExoI. We postulate that in the absence of TRF2$^{B}$, T-loops undergo branch migration to form double HJs, which then become substrates for cleavage by HJ resolvases, including MUS81, SLX1/SLX4, EMI1 and/or GEN1, leading to large telomere deletions and T-Circle formation[14,18,20,21,48].

### Telomere R-loops promote ultrabright telomere formation
R-loops are three stranded structures composed of DNA-RNA hybrid and a displaced ssDNA[85]. Failure to resolve persistent and detrimental

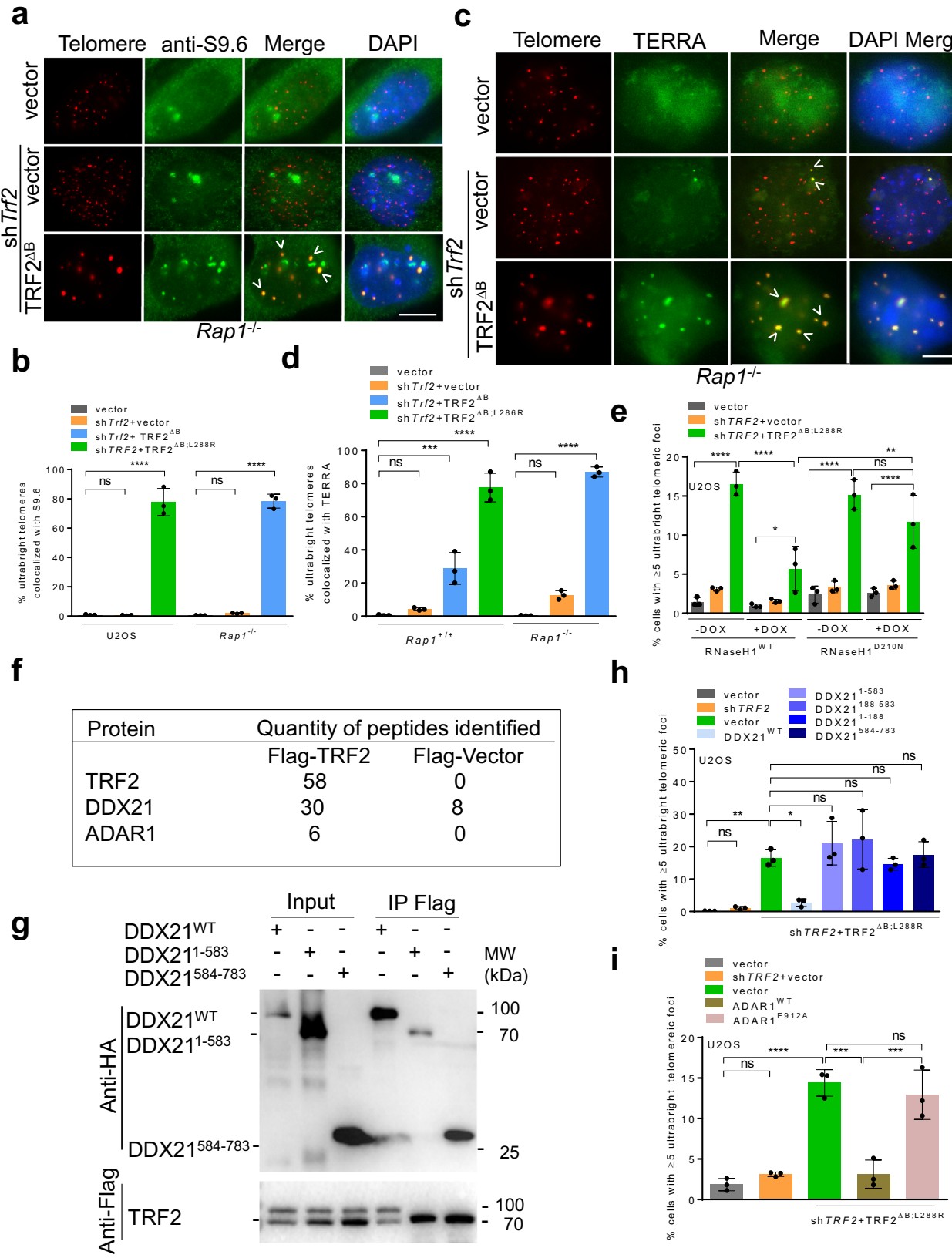

DNA-RNA hybrids impairs replication/transcription machineries[85–89]. R-loops are sources of replication stress and the ATR-CHK1 pathway is required to suppress their accumulation by recruiting R-loop resolving factors including DDX19 and SETX[90–93]. R-loop-mediated ATR activation also regulates MUS81 endonuclease activity to prevent excessive cleavage of R-loop-impeded replication forks[92]. Accumulation of R-loops at

telomeres has been shown to facilitate strand invasion[53,94]. Since replication stress enhances the formation of telomere bridging and UTs, it is likely that the activation of ATR-CHK1 in the absence of TRF2[B] and RAP1 recruits these R-loop resolving factors to telomeres in response to replication stress. Our data reveal that R-loop resolving proteins RNaseH1, DDX21 and ADAR1p110 repress R-loops formation at telomeres[51,57].

**Fig. 5 | TRF2$^B$ and RAP1 cooperate to repress TERRA and R-loop formation in ultrabright telomeres. a** IF-FISH showing cells containing ≥4 UTs colocalized with S9.6 antibody (green) in *Rap1*$^{-/-}$ MEFs cells treated with sh*Trf2* + TRF2$^{ΔB}$. Scale bars: 5 μm. **b** Quantification of percent UTs colocalized with S9.6 foci antibody in (**a**). Data represents the mean of three independent experiments ± SD from a minimum 300 nuclei analyzed per experiment. ****$P < 0.0001$ by one-way ANOVA. ns non-significant. **c** IF-FISH showing cells containing ≥4 UTs colocalized with TERRA (green) in *Rap1*$^{-/-}$ MEFs cells treated with sh*Trf2* + TRF2$^{ΔB}$. Scale bars: 5 μm. **d** Quantification of percent UTs colocalized with TERRA in (**c**). Data represents the mean of three independent experiments ± SD from a minimum 350 nuclei analyzed per experiment. ***$P = 0.0001$, ****$P < 0.0001$ by one-way ANOVA. ns non-significant. **e** Quantification of UT frequencies after doxycycline induced RNaseH1$^{WT}$ or catalytically dead RNaseH1$^{D210N}$ mutant in U2OS cells expressing indicated constructs. Data represents the mean of three independent experiments ± SD from a minimum of 350 nuclei analyzed per experiment. *$P = 0.0325$, **$P = 0.0035$, ****$P < 0.0001$ by one-way ANOVA. ns: non-significant. **f** IP-MS data using Flag-TRF2 and Flag-vector as baits showing R-loop associated proteins coprecipitated with Flag-WT TRF2 but not in Flag-vector in one representative experiment. **g** Co-IP with Flag antibody with lysates from 293 T cells expressing indicated proteins showing the interaction between TRF2 and DDX21 mutant depends on the DDX21 C-terminus. The blot shown is the representative of two independent experiments. **h** Quantification of UT frequencies in U2OS cells expressing Flag-DDX21$^{WT}$ and the indicated DDX21 deletion mutants in the presence of sh*TRF2* + TRF2$^{ΔB;L288R}$. Data represents the mean of three independent experiments ± SD from a minimum 300 nuclei analyzed per experiment. *$P = 0.0254$, **$P = 0.0064$ by one-way ANOVA. ns non-significant. **i** Quantification of UT frequency upon overexpression of GFP ADAR1p110 and catalytically inactive ADAR1p110$^{E912A}$ mutant in U2OS cells expressing indicated DNA constructs. Data represents the mean of three independent experiments ± SD from a minimum 350 nuclei analyzed per experiment. ***$P = 0.0001$, ****$P < 0.0001$ by one-way ANOVA. ns non-significant. Source data are provided as a Source Data file.

Since overexpression of RNaseH1 and ADAR1p110 but not their catalytic inactive mutants reduces the formation of telomere bridging and UT formation, our results suggest that TRF2$^B$ and RAP1 recruit these proteins to telomeres to resolve R-loops. This notion is further supported by increased abundance of TERRA at UTs in the absence of TRF2$^B$ and RAP1. In the absence of RAP1 and TRF2$^B$, increased accumulation of R-loops allows RPA and HDR factors to promote telomere-telomere bridging and the generation of UTs. Recently, replication stress mediated by loss of POT1 has also been shown to accumulate R-loops, highlighting the role of shelterin components in preventing the accumulation of this detrimental structure[95].

### TRF2$^B$ and RAP1 cooperate to repress RAD51-mediated HDR
Dysfunctional telomere-induced HDR is a key ALT activating mechanism[96]. UT formation in the absence of TRF2 and RAP1 is reminiscent of the telomeric clustering observed in ALT cells. While both RAD52-dependent and independent pathways are known to play a role in ALT activation[36,38,39,97], RAD51 has also been implicated in telomere clustering *via* inter-chromosomal homology and recombination between non-sister telomeres[27]. In addition, RAD51 catalyzes strand invasion, which is required for the formation of telomeric R-loops by TERRA[53]. Since both RPA and RAD51 localize to telomeric filaments prior to UT formation, it is likely that RAD51-mediated HDR is an early step in the generation of UTs. Indeed, we showed that knockdown of RAD51 but not RAD52 significantly reduced telomere bridging and UT formation. While telomere clustering and UTs have been reported mainly in ALT cell lines[24,96], our data reveal that absence of RAP1 and TRF2$^B$ promote the formation UTs not only in ALT proficient cells but also in ALT negative, telomerase positive cells. Consistent with this notion, we show that depletion of ATRX exacerbated telomere bridging and localization of PML bodies to UTs in RAP1 deficient, telomerase positive MEFs expressing TRF2$^{ΔB}$. Thus, both RAP1 and TRF2$^B$ protect telomeres from activating hallmarks of ALT by repressing RAD51-mediated HDR.

### 53BP1-mediated telomere mobility is required for formation of ultrabright telomeres
Previous reports reveal that DSBs and replication stress at telomeres increase telomere mobility to promote DNA repair[31]. Mobility of dysfunctional telomeres depends on the C-NHEJ factor 53BP1 and components of the LINC complex[29]. Interestingly, telomere mobility observed in ALT cells requires extensive end resection and a slew of HDR factors but is independent of 53BP1[27]. We show that filamentous telomere bridges and UT formation observed in the absence of RAP1 and TRF2$^B$ not only require HDR factors but also requires the N-terminal of 53BP1 to promote telomere mobility. In addition, UTs generated in the absence of RAP1 and TRF2$^B$ are much larger in size than those induced by TRF1-FokI in ALT cells, suggesting

distinctive mechanisms underlying telomere mobility in ALT proficient and ALT deficient telomerase positive cells.

### Lamin A represses the formation of ultrabright telomeres
Nuclear lamins are not only required for proper distribution of telomeres within nuclear spaces but also contribute to telomere homeostasis[61,66,98,99]. Lamin A has been shown to interact with TRF2 to promote the physical association of telomeres with interstitial chromatin to stabilize the chromosome-end structure[100]. Reconstitution of WT lamin A but not progerin reduces telomere bridging and the formation of UTs in the absence of RAP1 and TRF2$^B$, highlighting a role of lamin A in preventing inappropriate telomere mobility and telomere-telomere recombination. While we observed occasional large telomere foci in *Lmna*$^{-/-}$ MEFs, this phenotype was significantly exacerbated by the loss of RAP1 and TRF2$^B$. Unlike TRF1-FokI-mediated telomere clustering, which does not show telomeric foci localization to the nuclear periphery[27], we noticed that UTs prominently localized to nuclear periphery. This result supports previous data suggesting that lamin A plays a role in segregating mammalian telomeres to restrict aberrant telomere-telomere recombination[61,98,100]. We postulate that lamin A acts as a scaffold that restrains telomere movement in the nucleus, possibly through transient associations with RAP1. In support of this notion, we show the physical interaction between KU70/KU80 and RAP1$^{BRCT}$ is important to repress telomere bridging and UT formation. KU70 has been shown to interact with lamin A, but the function of these interactions is unknown[101]. Lamin A has also been shown to interact with SUN1 in the NE[102] while SUN1 physically interacts with RAP1 to connect the nuclear envelope to shelterin components[3].

Site-specific phosphorylation/dephosphorylation of lamin A is a major determinant of its ability to assemble and localize to the nucleus[103]. CDK1 dependent phosphorylation of lamin A on residue S392 inhibits assembly due to its increased nucleoplasmic localization[104,105]. Activation of CDK1 promotes nuclear lamin A disassembly with premature activation of the MUS81-SLX4 structure-specific endonuclease complexes, which promotes DNA breaks and formation of recombination intermediates[106,107]. Since loss of RAP1 and TRF2$^B$ promotes the activation of ATR-CHK1 pathway, we postulate that activation of these kinases impact phosphorylation-dependent lamin A disassembly. In support of this notion, we show that both CDK1 and ATR inhibitors abolish telomere bridging and UT formation. These results suggest that in the absence of RAP1 and TRF2$^B$ function, CDK1 and ATR-CHK1 kinases cooperate to restrict telomere-telomere recombination through phosphorylation of lamin A. Replication stress induced by POT1 loss has been previously shown to promote telomere localization to the nuclear pore, further supporting the role for shelterin components in regulating NE architecture and telomere homeostasis[108]. Taken together, our data suggest that lamin A interacts with TRF2 and RAP1 to maintain telomere homeostasis.

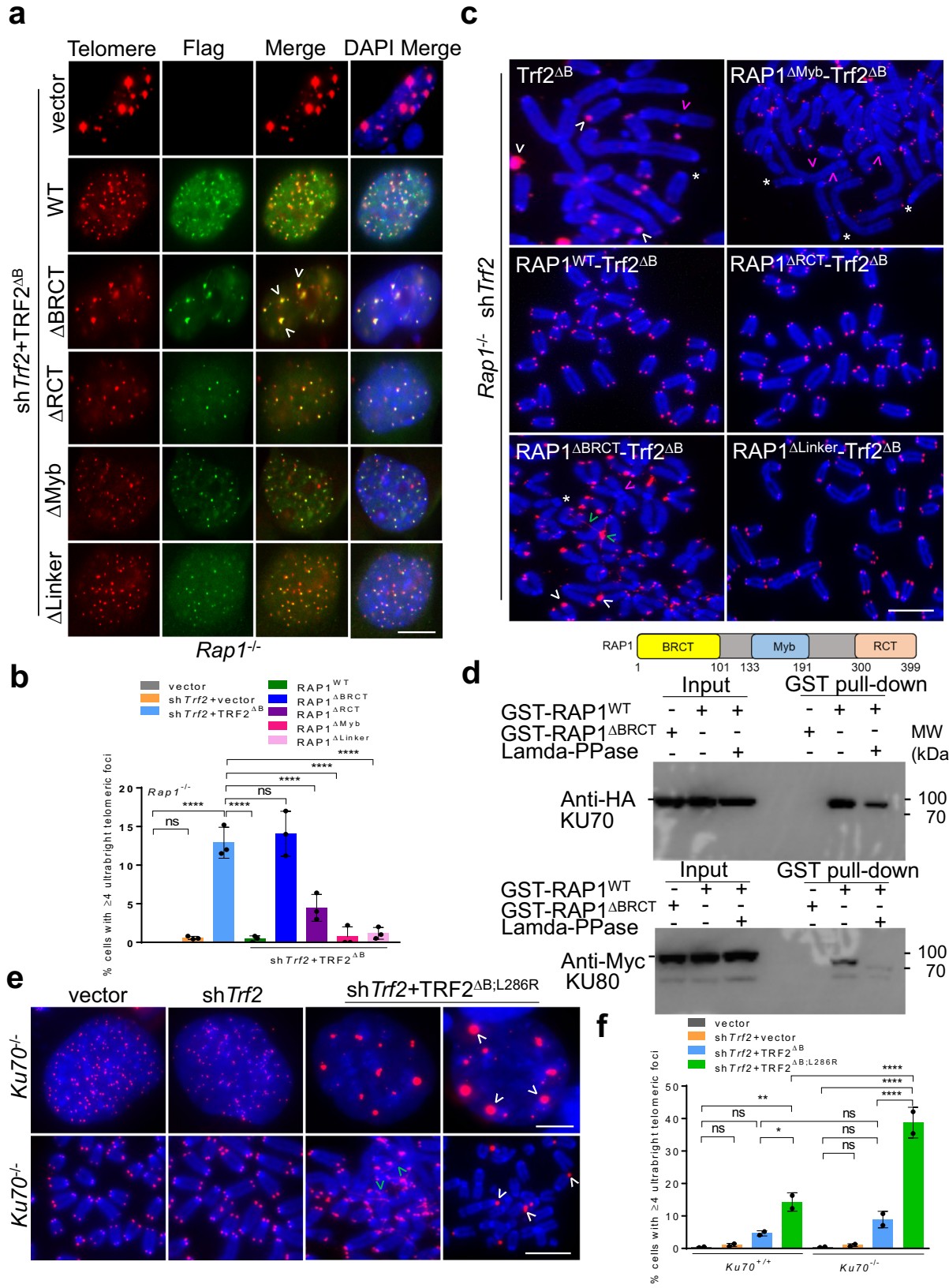

## Methods

### Cell Lines
293 T, IMR90, HeLa, *Rap1*[+/+], *Rap1*[-/-], *Ku70*[+/+], *Ku70*[-/-], *Nbs1*[-/-], *S3Bp1*[+/+], *S3Bp1*[-/-], *Lmna*[+/+] and *Lmna*[-/-] cells were cultured in DMEM supplemented with 10% FBS and maintained in 5% CO2 at 37 °C. Saos-2, U2OS and Dox inducible RNaseH1[WT] and RNaseH1[D2I0N] U2OS cells were

cultured in Macoy's medium. 1.0 μg/ml Doxycycline was used to induce RNaseH1[WT] and RNaseH1[D2I0N] expression.

### Antibodies and reagents
The antibodies used for western blot analysis and immuno-fluorescence were as follows: anti-CHK2 (BD Biosciences, #611570,

**Fig. 6 | RAP1 BRCT domain and KU70/KU80 cooperate to repress ultrabright telomeres. a** IF-FISH showing Flag-tagged RAP1-TRF2$^{\Delta B}$ fusion protein (green) localization at UTs in *Rap1*$^{-/-}$ MEFs expressing indicated constructs. Scale bars: 5 μm. **b** Quantification of UT frequencies in (**a**). Data shown as the mean of three independent experiments ± SD from a minimum 200 nuclei analyzed per experiment. ****$P$ < 0.0001 by one-way ANOVA. ns non-significant. **c** Representative images from three independent experiments of PNA-FISH on metaphase spreads showing CCCTAA-positive filaments (green arrowheads) and UTs (white arrowheads), signal free ends (*), fused chromosome without telomere at the fusion site (pink arrowheads) in *Rap1*$^{-/-}$ MEFs expressing Flag-tagged RAP1-TRF2$^{\Delta B}$ fusion proteins. At least 30 metaphases were analyzed per experiments. Scale bars: 15 μm.

**d** Purified GST-RAP1$^{WT}$ and GST-RAP1$^{\Delta BRCT}$ were subjected to a pull-down assay with 293 T cell lysates overexpressing HA-KU70 and Myc-KU80 in the presence and absence of lambda -PPase (protein phosphatase). The blot shown is the representative of two independent experiments. **e** PNA-FISH on interphase nuclei and metaphases showing the formation of CCCTAA-positive filaments (green arrowheads) and UTs (white arrowheads) in *Ku70*$^{-/-}$ *MEFs* expressing indicated constructs. Scale bars: 5 μm. **f** Quantification of UT frequencies in (**e**). Data shown as the mean of two independent experiments ±SD from a minimum 350 nuclei analyzed per experiment. *$P$ = 0.0277, **$P$ = 0.0030, ****$P$ < 0.0001 by one-way ANOVA. ns non-significant. Source data are provided as a Source Data file.

1:2000 dilution), anti phospho-CHK1 (Cell Signaling Technology, 133D3,#2348, 1:1,000 dilution), anti cGAS (D1D3G, Cell Signaling Technology, #15102, 1:1000 dilution), anti-SLX4 (H-39, # sc-135225, 1:1000 dilution), anti-RAD51 (H-92, # sc-8349, 1:1000 dilution), anti-RAD52 (F-7, #sc-365341, 1:1000), phosphorylated RPA32 (S33) (Bethyl, #A300-246A, 1:1000 dilution), anti-POLD1 (Bethyl, # A304-005A, dilution 1:1000), anti-SMARCAL1 (A2, #sc-376377, 1: 1000 dilution), anti-PML (H-238, #sc-5621, 1:1000), anti-Lamin A/C (E1, #sc-376248, 1:1000 dilution), anti-GFP (B2, #sc-9996, 1:2000 dilution) anti-DNA-RNA Hybrid (S9.6, Kerafast, #ENH001, 1:1000 dilution), anti-BrdU (B44, BD Biosciences, #347580, 1:1000 dilution), anti-ATRX (Abcam, #ab97508, 1:1000), anti-mTRF2 (gift from Karlseder Lab, Salk Institute, 1:1000 dilution), anti-TRF2 (4A794, Millipore, #05-521, 1:1000 dilution), anti-epitope tag antibodies anti-Flag (M2, #F3165, 1:2000 dilution) and anti-HA (HA-7, #H3663, 1:2000 dilution) from Sigma, anti-Myc (4A6, Millipore, #05-724, 1:2000 dilution). Anti-γ-Tubulin antibody (GTU-88, Sigma, #T6557, 1:5000 dilution) used for the internal control in western blots. Secondary antibodies for western blot: peroxidase-linked anti-mouse IgG (Amersham NXA931V, 1:5000 dilution), peroxidase-linked anti-rabbit IgG (Amersham NA934V, 1:5000 dilution). Secondary antibodies for immunostaining were purchased from Invitrogen and used at a 1:2000 dilution: Alexa Fluor 350 anti-rabbit (A11044), Alexa Fluor 350 anti-mouse (A11045), Alexa Fluor 488 anti-mouse (A11001), Alexa Fluor 568 anti-mouse (A11004), Alexa Fluor 488 anti-rabbit (A11008), Alexa Fluor 594 anti-rabbit (A11012). Aphidicolin (#A0781), Doxycycline (#D5027), Roscovitine (#557364) were purchased from Sigma. Ro-3306 (#S7747), CDK1 inhibitor purchased from Selleckchem. ATR inhibitors (AZD6738, VE-822, AZ20, BAY1895344) were gift from Bindra Lab at Yale. Anti-Flag M2 affinity gels (Sigma, #A2220) and Protein A/Protein G Sepharose beads (GE Healthcare, #17-6002-35) used to pull down overexpressed or endogenous proteins.

### Expression vectors and shRNAs
mTRF2WT, mTRF2$^{\Delta B}$, mTRF2$^{\Delta B;L286R}$ and Flag-mTRF2- RAP1 proteins tethered to TRF2$^{\Delta B}$ were cloned into pQCXIP-puro retroviral expression vectors. pLPC-Lamin A WT, pLPC-Progerin, pLPC-NMyc Myc-hTRF2$^{\Delta B}$, eGFP-TRF1 pWzl-Hygro, and pmGFP-ADAR1-p110, IF-GFP-ATRX, pMSCVpuro-eGFP-hcGAS, pMSCVpuro-eGFP-mcGAS from Addgene. All the constructs were confirmed by sequencing. Point mutations including hTRF2$^{\Delta B;L288R}$ were introduced using side-directed mutagenesis according to the manufacturer's protocol (Agilent QuickChange II XL Site-Directed Mutagenesis Kit, #500521). HA-53BP1 WT and mutants[109], HA-POLθ WT and mutants[35], Flag -TRF1-FokI WT and mutant[27], Fucci mKO-CTD1 and Fucci mAG-Geminin[25], CSII-EF-FLAG-ADAR1p110-WT-IRES-puro and mutant[57]. pcDNA3.1 HA-DDX21 and mutants from Dr. Yong Chen. pDest 3X GFP/HA Lamin WT including phosphomimetic or the phosphorylation deficient ATR and CDK1 mutants from Lilian Kabeche (Yale University). Mouse *Trf2* shRNA was used to deplete endogenous TRF2[109]. pRetroSuper sh*Rad51*, pRetroSuper sh*Ligase 3* were from Dr Madalena Tarsounas, Oxford, UK. shRNA against SLX4 was a gift from Dr Yie Liu, NIA. pGIPZ sh*Rad52* (V2LHS_171206, V3LHS_376616) from Ryan Jensen, Yale University. PlK.01 shlenti human *TRF2* shRNA (TRC N0000280026), shLenti *Exo1* (TRCN0000218614 TRCN0000238466 TRCN0000238468), shLenti *Atrx* (TRCN0000081910, TRCN0000 302073), shLenti *Claspin* (TRCN0000193573 TRCN0000175992 TRCN 0000193398), shLenti *Pcna* (TRCN0000294872 TRCN0000294805 TRCN0000287377), shLenti *Donson* (TRCN0000377075 TRCN0 000249773 TRCN0000201175), shLenti *Pold1* (TRCN0000071233 TRCN0000071234 TRCN0000071235) from Sigma. shLenti human *POLQ* and mouse *Polq* from Agnel Sfier, NYU School of Medicine.

### Retroviral infection in the cell lines.
DNA constructs were transfected into 293 T cells using Fugene 6 and packaged into retro or lentiviral particles. Viral supernatant was collected 48-72 h after transfection, filtered with 0.45 μm millex filter and directly used to infect immortalized MEFs or human cells.

### Western blot analysis.
For immunoblotting, trypsinized cells were lysed in urea lysis buffer (8 M urea, 50 mM Tris-HCl, pH 7.4, and 150 mM ß-mercaptoethanol). The lysates were denatured and then resolved on 4-12% SDS–PAGE gel. The separated proteins were then blotted on a PVDF membrane (Amersham Hybond, # 10600069), blocked with blocking solution (5% non-fat dry milk in PBS/0.1% Tween-20) for at least 1 h and incubated with appropriate primary antibody in blocking solution at least 2 h at room temperature (RT) or overnight at 4 °C. The membranes were washed 3×10 mins with PBS/0.1% Tween-20 and incubated with appropriate secondary antibody in blocking solution for 1 h at RT. Chemiluminescence western blot was performed using an ECL western Blotting Detection kit from Amersham (# RPN2232). ChemiDoc MP imaging system and Image Lab Touch software from BIO-RAD used to document the chemiluminescence western blots.

### Coimmunoprecipitation.
293 T or U2OS cells grown in 10 cm plates were co-transfected with epitope tag cDNAs or vector control. 48 h after transfection, cells were harvested and lysed in BCO 300 buffer (20 mM HEPES, pH 7.5, 10% glycerol, 1 mM EDTA, 0.1% (v/v) NP-40) and protease inhibitor. Supernatants were immunoprecipitated with appropriate endogenous or protein tagged antibody conjugated flag M2 affinity gels. Beads were washed thrice, and eluted proteins were analyzed by SDS-PAGE.

### Immunofluorescence and fluorescent in situ hybridization.
Cells grown on coverslips (Kemtech #0340-0150) were fixed for 10 min in 2% (w/v) sucrose and 2% (v/v) paraformaldehyde at RT followed by PBS washes. Coverslips were blocked in 0.2% (w/v) fish gelatin and 0.5% (w/v) BSA in PBS. Cells were incubated with primary antibodies and after PBS washes, cells were incubated with appropriate Alexa fluor secondary antibodies followed by washes in PBS + 0.1% Triton. For IF-FISH, the cells were further fixed with 4% (w/v) paraformaldehyde for 10 mins, followed by hybridization with TelC-Cy3 (CCCTAA)$_3$ PNA telomere probe (PNA Bio, F1002) in hybridization buffer (0.5 μg/ml tRNA, 1 mg/ml BSA, 0.06 × SSC, 70% formamide), denatured at 85 °C for 3 mins and then incubated at RT overnight in a humid chamber[109].

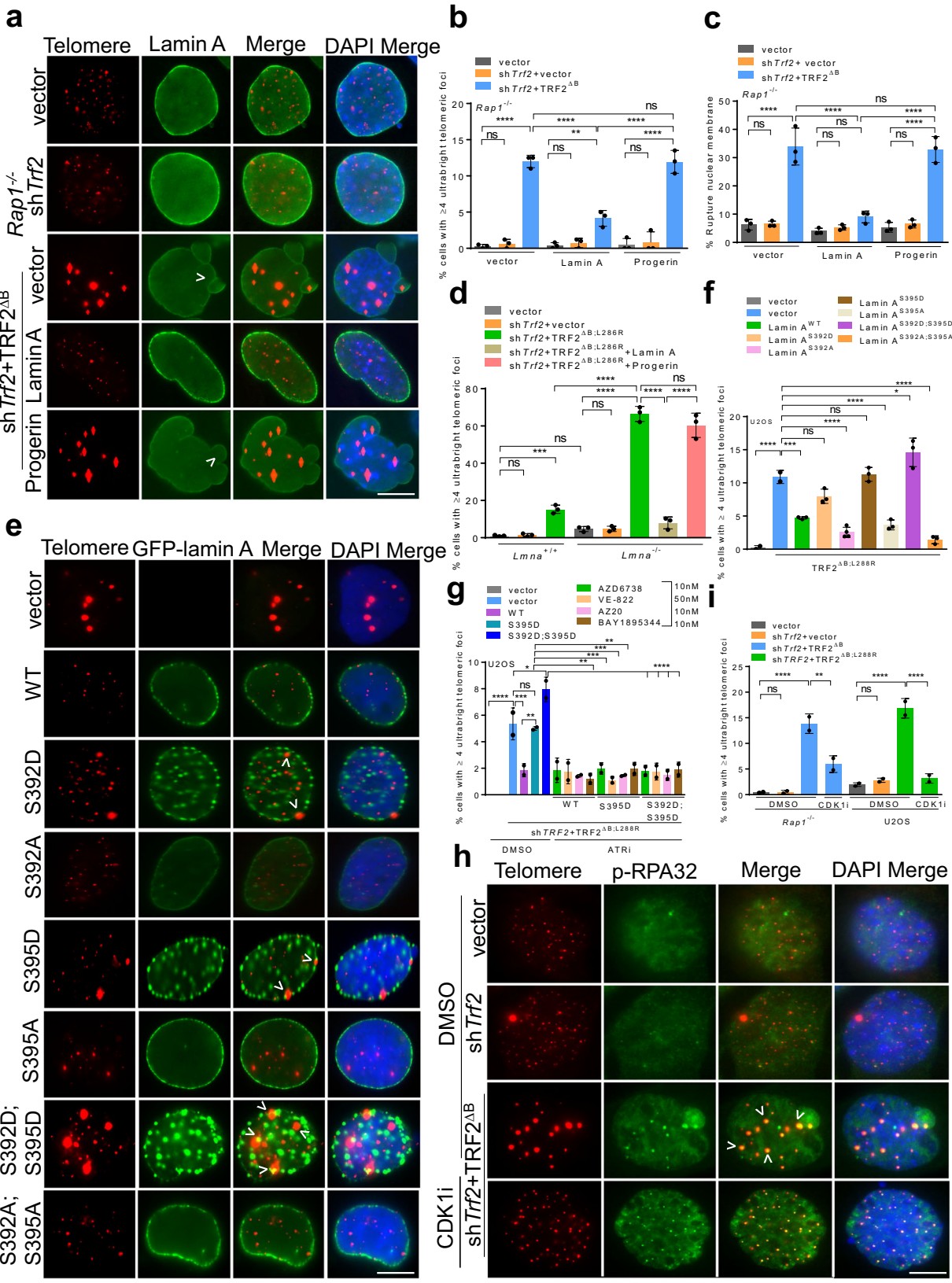

After washing the coverslips, DNA was stained with DAPI (Vectashield # H1200), and digital images captured at 10 ms or 100 ms exposure using NIS-Elements BR (Nikon) with a Nikon Eclipse 80i microscope and an Andor CCD camera. For EdU detection, cells labeled with 20 μM EdU for 30 mins were fixed as above followed by IF-FISH. After PNA FISH, cells were fixed with 4% PFA for 4 mins. Fixed cells were blocked with 3% BSA for 30 mins at RT followed by EdU detection according to manufacturer's protocol (Click-it EdU Alexa Fluor 488 imaging kit, Invitrogen #C10337).

**Live cell imaging.** To determine the telomere bridges and telomere size over time in live cells, cells expressing GFP-TRF1 grown on 6

**Fig. 7 | Defects in lamin A promote the formation of ultrabright telomeres. a** IF-FISH showing lamin A staining (green) in *Rap1*[−/−] MEFs expressing indicated constructs and upon expressing WT lamin A or Progerin. Scale bars: 5 µm. **b** Quantification of UT frequencies in (**a**). Data represents the mean of three independent experiments ± SD from a minimum 400 nuclei analyzed per experiment. **$P = 0.0036$, ****$P < 0.0001$ by one-way ANOVA. ns: non-significant. **c** Quantification of NE rupturing in cells expressing UTs in *Rap1*[−/−] MEFs expressing indicated constructs. Data represents the mean of three independent experiments ± SD from a minimum 400 nuclei analyzed per experiment. ****$P < 0.0001$ by one-way ANOVA. ns: non-significant. **d** Quantification of UT frequencies in *Lmna*[+/+] and *Lmna*[−/−] MEFs treated with sh*Trf2* + TRF2[ΔB;L286R]. Data represents the mean of three independent experiments ± SD from a minimum 300 nuclei analyzed per experiment. ***$P = 0.0008$, ****$P < 0.0001$ by one-way ANOVA. ns non-significant. **e** IF-FISH showing the impact of CDK1 and ATR phospho- and dephospho-GFP-tagged Lamin mutants (green) in the generation of UTs in U2OS cells expressing TRF2[ΔB;L288R]. Scale bars: 5 µm. **f** Quantification of UT frequencies in (**e**). Data represents the mean of three independent experiments ± SD from a minimum 350 nuclei analyzed per experiment. *$P = 0.0246$, ***$P = 0.0001$ ****$P < 0.0001$ by one-way ANOVA. ns non-significant. **g** Quantification of UT frequencies in U2OS cells expressing indicated constructs treated with ATR inhibitors (ATRi). Data represents the mean of two independent experiments ± SD from a minimum 300 nuclei analyzed per experiment. *$P = 0.0204$, **$P = 0.0031$, ***$P = 0.001$, ****$P < 0.0001$ by one-way ANOVA. ns non-significant. **h** Representative IF-FISH showing that 4.0 µM Ro-3306 CDK1 specific inhibitor (CDKi) reduced the formation of UTs but not p-RPA32(S33) (green) TIFs in *Rap1*[−/−] MEFs expressing indicated proteins. **i** Quantification of UTs in *Rap1*[−/−] MEFs and U2OS cells in the absence and presence of CDK1 inhibitor. Data represents the mean of two independent experiments ± SD from a minimum 300 nuclei analyzed per experiment. **$P = 0.0023$, ****$P < 0.0001$ by one-way ANOVA. ns non-significant. Scale bars: 5 µm. Source data are provided as a Source Data file.

microwell glass plate (Cellvis #P06-14-0-N) were imaged at different time interval using NIS-Elements BR (Nikon) with a Nikon Eclipse 80i microscope and an Andor CCD camera.

**Sub telomere FISH.** Methanol/Acetic acid fixed cells were spotted onto a glass microscope slide. Dried slides were immersed in 2x Saline Sodium Citrate (SSC) for 2 min at RT. Slides were then dehydrated in an ethanol series (70%, 85% and 100%), each for 2 min at RT. In total, 10 µl of sub telomere probe (CytoCell) were placed on the slides and then denatured the sample and probe simultaneously by heating the slide at 75 °C for 2 min. Slides were placed in a humid, lightproof container at 37 °C overnight. Slides were washed with 0.4x SSC + 0.3% NP40 for 2 min followed by wash in 2xSSC at RT for 2 min. After washing the slides, PNA-FISH was carried out using a TelC-Cy3 PNA telomere probe. DNA was stained with DAPI, and digital images were captured as described above.

**TERRA FISH.** Cells grown on coverslips were treated with cytobuffer (100 mM NaCl, 300 mM sucrose, 3 mM MgCl$_2$, 10 mM PIPES pH 7, 0.1% Triton X-100, 10 mM vanadyl ribonucleoside complex Sigma #R3380) for 7 min at 4 °C. Cells were rinsed briefly, fixed with 4% paraformaldehyde in PBS for 10 min at RT. Cells were then washed three times with PBS for 5 min each and then incubated with hybridization mix (10 nM TERRA FISH probe (TAACCC)7-Alexa488-3′, Integrated DNA Technologies, 50% formamide, 2× SSC, 2 mg/ml BSA, 10% dextran sulfate, 10 mM vanadyl ribonucleoside complex) for 18 h in a humidified chamber at 39 °C. Cells were washed with 2× SSC in 50% formamide three times at 39 °C for 5 min each, three times in 2× SSC at 39 °C for 5 min each, and finally once in 2× SSC at RT for 10 mins. Coverslips were than mounted on glass microscope slides with DAPI. Coverslips incubated with 200 µg/ml with RNaseA (Sigma #R6148), for 30 min at 37 °C before hybridization serves as negative control. Digital images were captured as described above.

**Chromosome analysis by telomere PNA-FISH and MiDAS.** Cells were treated with 0.5 mg/ml of Colcemid before harvest. Cells pelleted by centrifugation at 600 × g for 8 min were resuspended in 0.06 M KCl, incubated for 15 min at RT and washed three times with methanol:acetic acid (3:1 ratio). Metaphase spreads were prepared on microscope slides (Fisherbrand #22-038-100), treated with 0.5 mg/ ml of RNase A for 10 min at 37 °C and fixed with 3% formalin in PBS for 10 min at RT. PNA-FISH was performed as described above. After washing the slides, DNA was stained with DAPI, images were captured as described above. Ultrabright telomeres and telomeric bridges analyzed both in interphase nuclei and metaphase spreads. The percentage of telomere aberrations (UTs, telomere bridges, signal free ends, telomeric fusions with and without telomeres) observed is defined as: total number of telomere aberrations in 30−50 metaphase spreads analyzed divided by the total number of chromosomes examined ×

100%. MiDAS on metaphase spreads were performed as described[110]. Briefly, cells were labeled with 20 µM EdU for 1 h. After PNA FISH on metaphase spreads, slides were fixed with 4% PFA for 4 mins. Fixed cells were blocked with 3% BSA for 30 mins at RT followed by EdU detection according to manufacturer's protocol.

**Telomere length analysis and G-Strand overhang assays.** For in-gel detection of telomere length and G-stand overhang, a total of $1–2×10^6$ cells were suspended in PBS, mixed 1:1 with 1.8% pulse field agarose (BIO-RAD, # 1620137) in 1xPBS and cast into plugs. The plugs were then digested overnight at 50 °C with 20 mg/ml Proteinase K (Roche, #03115879001) in 10 mM sodium phosphate (pH 7.2) and 0.5 mM EDTA and 1% sodium lauryl sarcosine. DNA in plugs were subsequently digested by Hinf1/Rsa1 overnight at 37 °C. The next morning, plugs were washed once with 1xTE and equilibrated with 0.5xTBE. Exo 1 digestion in the plugs performed with 100U Exo I at 37 °C overnight or T7 endonuclease I treatment with

40U at 37 °C for 1 h. The plugs were loaded onto a 0.8% agarose gel in 0.5xTBE and run on a CHEF-DRII pulse field electrophoresis apparatus (BIO-RAD). The electrophoresis parameters were as follows: Initial pulse: 0.3 s, final pulse: 16 s, voltage: 6 V cm[-1] run time: 14 h. Dried gel pre-hybridized with Church mix for 2 h at 55 °C and hybridized overnight at 55 °C in Church mix with telomere repeat oligonucleotide probe $^{32}$P-labeled-(CCCTAA)$_4$, [γ−$^{32}$P ATP #BLU502H250UC]. After hybridization, the gel was washed three times for 20 min with 4xSSC/0.1% SDS at 37 °C, thrice with 4xSSC/0.1% SDS at 55 °C and exposed to a phosphoimager screen overnight. After exposure, the screen was scanned on a Typhoon Trio imager system and ImageQuant TL software (GE). The gels were subsequently denatured and hybridized using the same probe.

**2D-TRF Southern assay.** Thirty micrograms of total genomic DNA digested with *Hinf*1/*Rsa*1 was resolved in the first dimension in 0.4% agarose gel in 0.5xTBE at 26 V for 15 h. Ethidium bromide-stained gel slices were rotated 90° from the first gel and cast in a 1% agarose gel and resolved at 115 V for 4 h. Dried gel were denatured and subsequently hybridized as described above. For T-complex, Hinf1/Rsa1 digested DNA were treated with 40 units of T7 endonuclease I at 37 °C for 1h. After exposure, hybridization signals were analyzed with a Typhoon Trio imager system and ImageQuant TL software.

**T-circle and C-circle assay.** T-circle assay was performed with 3.0 µg genomic DNA digested with *Alu*I and *Hinf*I and then annealed with 10 µM Thio-(C$_3$TA$_2$)$_3$. Rolling circle amplification was performed in the presence and absence of Phi29 DNA polymerase[111]. Amplified products were subjected to Southern blotting and hybridized with $^{32}$P-labeled T$_2$AG$_3$ oligonucleotides. C-circle assay was performed with 400 ng genomic DNA digested with *Alu*I and *Hinf*I. Rolling circle amplification was performed in the presence and absence of 1.0 unit

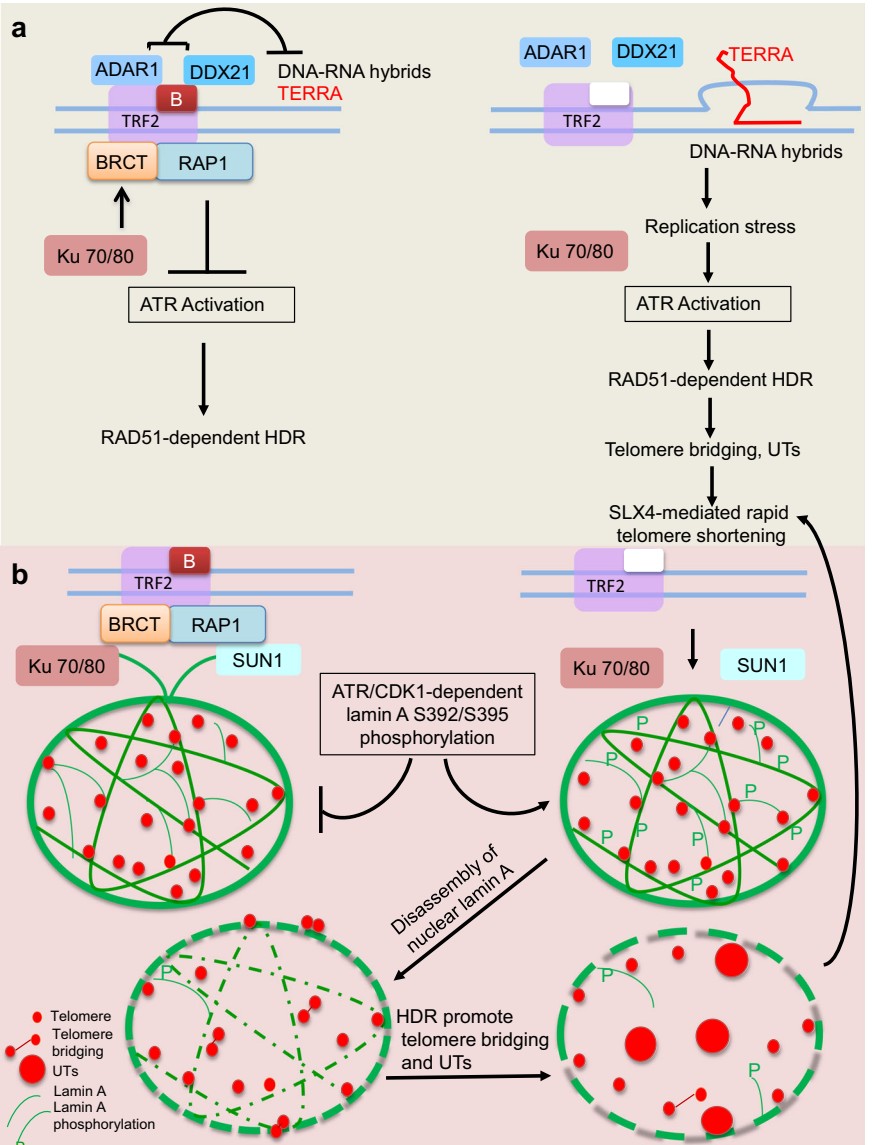

**Fig. 8 | Speculative schematic diagram illustrating the formation of ultrabright telomeres in the absence of RAP1 and TRF2B. a** In WT cells, TRF2B and RAP1 cooperate to protect chromosome ends from inappropriate activation of ATR-dependent HDR. TRF2B recruits DDX21 and ADAR1p110 to telomeres to resolve DNA-RNA hybrids, preventing the formation of telomeric R-loops, TERRA and UTs. In the absence of RAP1 and TRF2B, stalled replication forks lead to activation of ATR-CHK1, accumulation of telomeric R-loops and TERRA, promoting UT formation. Unresolved telomere recombination intermediates become substrates for cleavage by the SLX4 endonuclease, leading to catastrophic telomere shortening *via* T-loop HDR and the formation of telomere-free chromosome fusions. **b** Lamin A acts as a scaffold that restrains telomere movement in the nucleus, possibly through transient associations of RAP1 with KU70/KU80 and SUN1, known NE binding proteins. In the absence of RAP1 and TRF2B, ATR-CHK1 and CDK1 dependent phosphorylation of lamin A results in the disassembly of nuclear lamin A. Decompartmentalization of telomeres due to disruption of the lamin A architecture facilitates HDR, telomere-telomere clustering and UT formation.

Phi29 DNA[42]. Amplified C-circles products were subjected to dot blot and hybridized with ³²P-labeled-(CCCTAA)₄ oligonucleotides. Southern blot and dot blot images were scanned Typhoon Trio imager system and ImageQuant TL software. The level of ³²P incorporation in the Phi29 negative control samples was subtracted from the samples that contained the Phi29 DNA polymerase.

### Reporting summary
Further information on research design is available in the Nature Portfolio Reporting Summary linked to this article.

## Data availability
All the data supporting the findings in this study are available within the paper and its supplementary data files. Source data are provided with this paper. phospho ELM, a database of S/T/Y phosphorylation sites is available at http://phospho.elm.eu.org/. Source data are provided with this paper.

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

## Acknowledgements

We would like to thank all the members of the Chang lab for helpful
suggestions. We thank Dr. Susana Gonzalo (St Louis University School of
Medicine) for kindly providing *Lmna*<sup>+/+</sup> and *Lmna*<sup>−/−</sup> MEFs, Dr. Maria
Blasco (CNIO Spain) for providing *Rap1*<sup>+/+</sup> and Rap*1*<sup>−/−</sup> MEFs, Dr. Lilian
Kabeche (Yale University) for Dox inducible RNaseH1<sup>WT</sup>, RNaseH1<sup>D210N</sup>
U2OS cells and ATR/CHK1 phosho and dephospho Lamin A mutants, Dr.
Ranjit Bindra (Yale University) for ATR inhibitors and Dr. Li Peining (Yale
University) for subtelomeric chromosome FISH probes. We

acknowledge CISF staff Dr. Kamalakar Ambadipudi in the YCC CCSG
Shared Resources for training and assistance on the use of Cesium
Irradiator Shared Resource. T.S. and J. M. were both generously sup-
ported by the Hsieh Bylinsky Cancer Fellowship. Y.C. was supported by
grants from Strategic Priority Research Program of the Chinese Acad-
emy of Sciences (XDB37010303). S.C. is supported by the Department
of Defense grants W81XWH191005, BC210086 and NIH grants
RO1CA202816 and RO1GM141350.

## Author contributions

R.R. and S.C. conceived the project and designed the experiments; K.B.
and R.R. performed the biochemistry and molecular biology experi-
ments; W.S., X.Y., X.L. and Y.C. performed DDX21 and RAP1BRCT protein
interaction experiments; A.A.H performed live imaging and time
dependent UTs generation experiments; J.M. performed ADAR1
immuno-FISH; T.S. performed Western blotting for POLD1 depletion and
ATRX reconstitution experiments; K.B, R.R, Y.C. generated data for the
figures. R.R. Y.C., and S.C. analyzed and interpreted the data, composed
the figures, and wrote the paper.

## Competing interests

The authors declare no competing interests.

## Additional information

**Supplementary information** The online version contains
supplementary material available at

Rekha Rai or Sandy Chang.

**Peer review information** *Nature Communications* thanks the anon-
ymous reviewers for their contribution to the peer review of this
work. Peer reviewer reports are available.

