## [Peer Review File · Nature Communications]

Homology directed telomere clustering, ultrabright telomere formation and nuclear envelope rupture in cells lacking TRF2B and RAP1REVIEWER COMMENTS

Reviewer #1 (Remarks to the Author):

The authors build on previous publications to further explore the mechanism by which RAP1 loss and TRF2-basic domain deletion leads to homology-directed telomere loss and ultrabright telomere (UT) formation. They show mechanistically that disruption of RAP1 interaction with Ku70/80, via its BRCT domain, disruption of Lamin A interaction via Ku70/80 loss, and disruption of Lamin A phosphorylation by CDK1 all promote UT formation. Additionally, they show that R-loops and 53BP1 mobility are required for telomere filament and UT formation.

A key observation shown in Figure 1 is the transition of the telomere-bridge to UT phenotype & fused telomere ends at 72-96 h, then becoming telomere-free fused ends at 120 h. Further, it is suggested in Figure 4 that UTs are derived from replication intermediates that become entangled, supported by observed inter-chromosomal telomere bridges during metaphase. It is not clear whether UTs are also fused telomeres on chromosome ends or ECTRs before signal-free telomere fusion. It is likely that ECTRs are the result of excision of the recombinant structure. Do the authors see a decrease in t-complex and an increase in t-circle (via time-point assaying) when approaching 120 h via 2D-TRF or by IF? Overall, this is an interesting study, evaluating telomere dynamics and the regulators of replication stress/UT formation at telomeres. The data are strong and well presented; however, many of the observations stem from previous and well-established findings by this group and others, somewhat detracting from the novelty. In addition, the paper would benefit from a more coherent story connecting the DNA-RNA hybrids/replication stress, KU70/KU80, Lamin A/nuclear envelope observations. This would substantially improve the comprehension and impact of the manuscript.

Specific comments

- Since the UT foci are not retained according to Figure 1, it is not clear but implied that they become ECTRs? This should be visualized by microscopy or time-course 2D-TRFs.
- EdU/RPA staining should be performed for Figure 1 to further support that they are replication intermediates?
- It would be useful to determine whether other BIR components (eg TOPBP1 or POLD) suppress the UT phenotype with the TRF2 Δ -B RAP1 $^{-/-}$ background.
- For multiple figures, representative panels should be adequately labelled for one channel of interest eg red is PNA-FISH in Figure 1. Scale bars are missing from images throughout.
- Error bars for bar graphs are at times are confusing (2C, 2F, 3A, 3D). Having staggered flat lines when comparing the same column multiple times would aid visualisation.
- What are the left versus right panels of the 2D-TRFs?
- Figure 5: TERRA-FISH should be accompanied by an RNaseH control to demonstrate specificity.
- Figure 7: Claim of CDK1-dependency ideally should be proven with RO 3306 not Roscovitine which inhibits multiple CDKs.

Reviewer #2 (Remarks to the Author):

Summary

This manuscript from Rai et al investigates the mechanism by which the shelterin proteins TRF2 and RAP1 protect telomeres from homology directed repair (HDR). The focus is on the basic domain of TRF2. The authors observe that telomeres lacking the basic domain of TRF2 and RAP1 form ultrabright telomeres that exhibit evidence of multiple participating chromosomes and possess classic markers of HDR. Staining with the S9.6 antibody suggests the presence of RNA-DNA hybrids inside the ultrabright telomeres. Consistent with these results, proteomic analysis reveals that TRF2 interacts with the RNA processing proteins DDX21 and ADARp110. RAP1 appears to repress telomeric recombination and ultrabright telomere formation by recruitment of the Ku complex. Finally, the authors show that

expression of lamin A mutants induces nuclear envelope defects and ultrabright telomere formation. Overall, the experiments presented here are generally rigorous and well-performed. The authors make a strong case that ultrabright telomeres represent dysfunction caused by HDR at telomeres and that the basic domain of TRF2 and RAP1 play essential roles in blocking this process.

The manuscript presents a large amount of data and it is often difficult to see how specific segments of the story fit together i.e. the relative importance of TRF2-interacting partners, RAP1-interacting proteins, and the nuclear envelope in suppressing HDR at telomeres. It is also often unclear if specific observations necessarily tie together. For example, TRF2 interactions with ADAR1 and DDX21 are demonstrated and overexpression of these proteins affects ultrabright telomere formation. However, how the interaction with TRF2 affects DDX21 and ADAR1 function at telomeres is not demonstrated. In addition, the connections between telomere dysfunction and nuclear envelope defects are not clear from the data presented.

Specific points

The S9.6 antibody has the potential for creating pervasive artifacts when imaging RNA:DNA hybrids (PMID: 33830170). It is therefore important to perform RNaseH overexpression controls to verify interpretation of the anti-S9.6 immunofluorescence results in Fig. 5A,B. I realize the experiment may be difficult since RNaseH1 overexpression decreases the number of ultrabright telomere foci (Fig. 5E), but there appear to be a sufficient number of cells to allow for this critical analysis.

The roles of DDX21 and ADAR1 in repressing ultrabright telomere formation appear clear from the data. However, a role for the TRF2 interaction in facilitating their function is not investigated. I suggest the authors disrupt the TRF2-DDX21 and/or TRF2-ADAR1 interactions with targeted mutations to more directly test the role of TRF2 in facilitating DDX21 and ADAR1 function at telomeres.

Gaps in lamin A signal intensity at the nuclear envelope do not necessarily demonstrate a loss of compartmentalization as suggested by the authors (Fig. S6C,D). The authors should support this analysis by live-cell, time-lapse imaging of NLS-GFP reporter (PMID: 22567193). Loss of this marker is required to demonstrate a loss of compartmentalization. Other commonly used surrogate for nuclear envelope rupturing are localization of the DNA-binding proteins BAF and cGAS to the size of nuclear envelope rupture.

The authors focus a large portion of Fig. 7 on understanding how ATR-mediated phosphorylation affects Lamin A function. Much of this is based on unpublished data showing that ATR-mediated phosphorylation of Lamin A promotes micronucleus rupturing. Without the unpublished data it is very difficult to understand the rationale for promoting these experiments. I suggest the authors supply the unpublished data, cite the appropriate reference when appropriate, or remove this portion of Fig. 7. Related to the point above: It is not clear why ATR phosphorylates Lamin A. Is this an aberrant ATR target induced by loss of RAP1 or the TRF2 basic domain?

Reviewer #3 (Remarks to the Author):

R. Rai, S. Chang and colleagues have published in 2016 that Rap1 and the basic domain of TRF2 inhibit HDR at telomeres, thereby preventing recombination-mediated telomere fusions and deletions. In this manuscript, the same lead authors investigate the mechanisms involved in these recombination events. First, they show that the initiating events of telomere recombination in the absence of Rap1 and TRF2-B is telomere bridging events, which are mediated by 53BP1 and recruit RPA and RAD51. Telomere bridging, via RAD51, further mediate telomere clustering into ultrabright telomere foci (UTs), which are also promoted by SLX4 and inhibited by the helicase domain of Pol Theta. UTs are reminiscent of ALT-associated PML bodies: they colocalize with PML and are enlarged upon loss of ATRX. In the absence of Rap1 and the basic domain of TRF2, telomeres are prone to replication stress due to increased TERRA-telomere R-loops, leading to unresolved recombination intermediates. This replication stress is alleviated by several proteins, including SMARCAL1 and the CPD complex, as well as DDX21 and ADARp110, which directly interact with TRF2. The authors then show that the BRCT

domain of Rap1 is responsible for preventing these recombination events. They find that RAP1-BRCT interacts with phosphorylated Ku70/80, and that UTs are increased in Ku negative cells. Finally, they find that Lamin A inhibits telomere bridging and clustering. In cells lacking RAP1 and TRF2-B, activation of ATR and CDK1 induces Lamin A phosphorylation, disassembly and therefore induction of telomere bridging.

The presented data are compelling and support the claims of the authors. The findings are numerous and novel. Many of the findings here would benefit from a more in-depth mechanistic analysis, which I understand is not feasible due to the large number of molecular processes analyzed here. Yet, a more compelling final model summarizing all the findings should be included.

In conclusion, I believe that the data presented here significantly advance our understanding of how shelterin protects telomeres, and are of broad interest to the readers of Nature Communications. I will suggest only experiments that I believe are critical prior to publication.

1. In general, the observed phenotypes are reminiscent of ALT features.

- Unfortunately, the experiments are either performed in ALT cells or in MEFs, in which telomeres are very long and display singular chromatin. Therefore, some key finding must be reproduced in non-ALT human cells: colocalization of UTs with PML, SLX4, SMARCAL1
- To which scale are the phenotype observed ALT-like? Authors should provide C-circle and MIDAS analysis in non-ALT cells
- Does re-expression of ATRX reduce UTs in ALT cells?
- Is TERRA increased or just R-loops?

2. The Ku70/80 phenotype is puzzling. If RAP1-mediated recruitment Ku70/80 at telomeres mediates the Ku-dependent inhibition of HRD, then suppression of Ku in Rap1-/- or TRF2-L286R cells should not have any effect. The data suggest a redundant recruitment of Ku, for which TRF2 have been suggested to play a role. To what extent is Ku70/80 lost at telomeres upon Rap1 suppression? This point should be at least discussed.

3. As stated above, a compressive model including all the findings is needed to grasp and synthesize all the results presented here. The current model mostly focuses on Lamin A.

4. In several instances, the text description does not correctly reflect the results. This should be corrected.

- In page 5, the authors state that UT formation "declined by 120h". This is true for fig 1C but not Supp 1B-C. The text should be more specific.
- In page 6, it is stated that depletion of RAD51 decreased formation of UTs by 3-fold, yet there is no quantification.
- Twice in page 6, in pages 8 and 9, it is question of "100%" co-localization, which is not correct. Replace by "nearly 100%".
- In page 7, it is stated that TRF1-FOK1 results in an increase in average telomere size in U2OS and MEFs. However, in MEFs, the differences are not statistically significant.

REVIEWER COMMENTS

Reviewer #1 (Remarks to the Author):

We thank this reviewer for stating that “the data are strong and well presented.”

Many of the observations stem from previous and well-established findings by this group and others, somewhat detracting from the novelty.

We respectfully disagree with this statement. Although we have previously shown that deletion of the TRF2 basic domain, coupled with RAP1 loss results in increased HDR, in this report we show the following novel findings:

- *Loss of TRF2^B and RAP1 promotes the formation of ultrabright telomeres (UTs)*
- *UT formation is due to telomere clustering and formation of DNA-RNA hybrids*
- *TRF2^B and RAP1 recruit DNA-RNA modulating proteins DDX21 and ADAR1p110 to repress UT formation*
- *UT formation depends on the integrity of the nuclear envelope (NE)*

In addition, the paper would benefit from a more coherent story connecting the DNA-RNA hybrids/replication stress, KU70/KU80, Lamin A/nuclear envelope observations. This would substantially improve the comprehension and impact of the manuscript.

We have modified the manuscript and include a new figure (Reviewer Figure 14) to make the link between TRF2-RAP1, formation of DNA-RNA hybrids to the NE clear.

Specific comments:

Since the UT foci are not retained according to Figure 1, it is not clear but implied that they become ECTRs? This should be visualized by microscopy or time-course 2D-TRFs.

We thank the reviewer for raising this interesting question. Our time course experiments revealed that the number of UTs formed peaked at ~96h and declined by 120h (Fig. 1C). However, the decline in the number of UTs observed directly correlated with an increase in UT size (measured as area of UT focal signal) due to increased HDR (Suppl. Fig. B, C). We now present new data (Reviewer Fig. 1) to support the notion that this increased UT size is due to increased telomere recombination and T-complex formation. 2D gel electrophoresis revealed that T-complexes increased in Rap1^{-/-} MEFs expressing TRF2^{AB} in a time-dependent manner (Reviewer Figs. 1A, 1B). In addition, we also see an increase in the number of telomere circles (TCs) formed at 120 hrs (Reviewer Fig. 1A). In addition, a TC assay revealed that increased telomere recombination indeed promoted the accumulation of TCs in a time dependent manner (Reviewer Fig. 1C). This data is now incorporated in the manuscript as Supp. Figs. 4E-4G.

A

B

C

Reviewer Fig. 1. (A). T-complex and TCs analysis using two-dimensional (2D) gel electrophoresis from genomic DNA isolated from *Rap1*^{-/-} MEFs expressing TRF2^{ΔB} at the indicated time points. The gel was denatured, dried, and hybridized in situ with a radiolabeled (CCCTAA)₄ oligonucleotide probe to detect T-complex (outlined by the black box), telomere circles (green arrowheads), single-stranded telomere DNA (red arrowheads), double-stranded telomere DNA (blue arrowheads). (B). Quantification of T-complex in (A). T-complex signal intensities in vector was set at 100%. (C). T-circle assay showing ECTRs at indicated timepoint in *Rap1*^{-/-} MEFs expressing TRF2^{ΔB}. Genomic DNA isolated from indicated treatment subjected to T-circle amplification in the presence and absence of phi29 DNA polymerase. Phi29 dependent amplification and linear telomere restriction fragments (TRFs) products of TCs were detected by in-gel hybridization using ³²P-labeled-T₂AG₃ telomeric probes.

EdU/RPA staining should be performed for Figure 1 to further support that they are replication intermediates?

We agree with the reviewer and now we present new data showing the colocalization of EdU with p-RPA 32 (S33) in *Rap1*^{-/-} MEFs expressing TRF2^{ΔB}, suggesting that UTs are the product of replication intermediates (Reviewer Figs. 2A, 2B). This data is now displayed in the manuscript as Supp. Figs. 4B, 4C.

Reviewer Fig 2. (A). Representative IF-FISH images showing the localization of telomeres with EdU (green) containing p-RPA32 (S33) (blue) in *Rap1*^{-/-} MEFs expressing indicated constructs. Telomeres were detected with a Cy3-OO-(CCCTAA)₄ PNA probe (red). (B). Quantification of percent telomeres colocalized with EdU containing p-RPA32 (S33). Data represents the average of three independent experiments ± SD from a minimum 200 nuclei analyzed per experiment. ****P<0.0001 by one-way ANOVA. ns: non-significant.

It would be useful to determine whether other BIR components (eg TOPBP1 or POLD) suppress the UT phenotype with the TRF2delta-B RAP1^{-/-} background.

We have shown that shRNA mediated knockdown of RAD52 in U2OS cells expressing TRF2^{ΔB:L288R} did not impact the formation of UTs, suggesting that BIR is likely not involved in UT formation (Fig. 3F, Supp. Fig. 3D). To address whether other BIR components are involved in the generation of UTs, we show new data where we knocked down POLD1, the catalytic subunit of POLD, using three independent shRNA (Reviewer Fig. 3A). Surprisingly, depletion of POLD1 reduced the number of p-RPA32 (S33) positive TIFs and the number of UTs in *Rap1*^{-/-} MEFs expressing TRF2^{ΔB} (Reviewer Fig. 3B-3D) suggesting a role for POLD1 in the formation of UTs. These finding are consistent with the role of POLD subunits in HDR^{1,2}. We now include this figure as Supp. Figs. 3E-3H in the manuscript.

Reviewer Fig. 3. (A). Immunoblot showing knockdown efficiency of *Pold1* with three different shRNAs in *Rap1*^{-/-} MEFs expressing indicated DNAs. Anti-POLD1 and TRF2 antibody used to detect POLD1 and TRF2 proteins. γ -Tubulin served as loading control. (B). Representative IF-FISH images showing that depletion of POLD1 reduced the formation of UTs (red) and p-RPA32 (S33) TIFs (green) in *Rap1*^{-/-} cells treated with shTrf2 and expressing TRF2^{ΔB}. Telomeres were detected with a Cy3-OO-(CCCTAA)₄ PNA probe (red) and 4,6-diamidino-2-phenylindole (DAPI) was used to stain nuclei (blue). (C). Quantification of UTs in (B). Data represents the average of two independent experiments \pm SD from a minimum 350 nuclei analyzed per experiment. ***P*=0.001 by one-way ANOVA. ns: non-significant. (D). Quantification of percent UTs colocalized with p-RPA32 (S33) in (B). *****P*<0.0001 by one-way ANOVA.

For multiple figures, representative panels should be adequately labelled for one channel of interest eg red is PNA-FISH in Figure 1. Scale bars are missing from images throughout.

We have provided the color channels and scale bars in each figure.

Error bars for bar graphs are at times are confusing (2C, 2F, 3A, 3D). Having staggered flat lines when comparing the same column multiple times would aid visualization.

We agree with this point and now Figs. 2C, 2F, 3A, 3D have been modified.

What are the left versus right panels of the 2D-TRFs?

The left and right panels in 2Ds-TRFs are +T7 endonuclease I and -T7 endonuclease I control treatments. We now made this clear in the 2D-TRF displayed as Fig. 4G.

Figure 5: TERRA-FISH should be accompanied by an RNaseH control to demonstrate specificity.

We agree with the reviewer and show new data that treatment with RNaseH dramatically reduced TERRA foci formation, demonstrating that TERRA foci is composed of RNA-DNA hybrids (Reviewer Figs. 4A, 4B). This figure is now shown as Supp. Figs. 5A, 5B in the manuscript.

Reviewer Fig. 4. (A). Representative images of IF-FISH showing the loss of TERRA signal (green) after incubating the fixed cells with 5U RNaseH for 2h. (B). Quantification of TERRA foci in (A). Data obtained from two independent experiments \pm SD from a minimum 250 nuclei analyzed per experiment. * $P=0.028$ by T test.

Figure 7: Claim of CDK1-dependency ideally should be proven with RO 3306 not Roscovitine which inhibits multiple CDKs.

We agree with this reviewer and present new data showing that the CDK1 specific inhibitor RO 3306 significantly reduced the formation of UTs but not p-RPA32 (S33) in both Rap1^{-/-} MEFs and U2OS cells expressing TRF2^{AB} or TRF2^{AB:L288R}, respectively (Reviewer Figs. 5A-5E). Activation of p-RPA32 (S33) by RO 3306 is consistent with suggesting that RO 3306 activates DNA damage responses in cells with the recruitment of DDR markers including p-RPA32³ (Reviewer Figs. 5D, 5E). Our data suggest that CDK1 dependent phosphorylation of Lamin A^{S392} promotes Lamin A disassembly and UT formation. We now include Reviewer Figs. 5A and 5C in the manuscript as Figs. 7H, 7I.

Reviewer Fig. 5. (A). Representative IF-FISH showing that 4.0 μ M CDK1 inhibitor RO 3306 reduced the formation of UTs but not p-RPA32 (S33) TIFs in Rap1^{-/-} MEFs expressing the indicated proteins. (B). Representative IF-FISH showing that 4.0 μ M CDK1 specific inhibitor RO 3306 reduced the formation of UTs but not p-RPA32 (S33) TIFs in U2OS cells expressing indicated proteins. (C). Quantification of UTs in Rap1^{-/-} MEFs and U2OS cells in the absence or presence of CDK1 inhibitor RO 3306. Data represents the average of two independent experiments \pm SD from a minimum 300 nuclei analyzed per experiment. ** $P=0.002$, **** $P<0.0001$ by one-way ANOVA. ns: non-significant. (D). Quantification of p-RPA32 (S33) colocalizing with telomeres in the absence and presence of CDK1 inhibitor RO 3306. *** $P=0.0003$ by one-way ANOVA. ns: non-significant. (E). Immunoblot showing the effect of CDK1 inhibition on the activation of p-CHK1 and p-RPA32 (S33) in Rap1^{-/-} MEFs expressing indicated proteins. γ -Tubulin served as loading control.

Reviewer #2 (Remarks to the Author):

We thank this reviewer for stating that “overall, the experiments presented here are generally rigorous and well-performed. The authors make a strong case that ultrabright telomeres represent dysfunction

caused by HDR at telomeres and that the basic domain of TRF2 and RAP1 play essential roles in blocking this process.”

The manuscript presents a large amount of data and it is often difficult to see how specific segments of the story fit together i.e. the relative importance of TRF2-interacting partners, RAP1-interacting proteins, and the nuclear envelope in suppressing HDR at telomeres.

We now show Reviewer Fig. 14 (Supp. Fig. 8 in the manuscript) linking these pathways together in a coherent manner.

Specific points

The S9.6 antibody has the potential for creating pervasive artifacts when imaging RNA:DNA hybrids (PMID: 33830170). It is therefore important to perform RNaseH overexpression controls to verify interpretation of the anti-S9.6 immunofluorescence results in Fig. 5A,B. I realize the experiment may be difficult since RNaseH1 overexpression decreases the number of ultrabright telomere foci (Fig. 5E), but there appear to be a sufficient number of cells to allow for this critical analysis.

We agree with the points raised by the reviewer. We now show new data that inducible expression of *RNaseH1^{WT}* but not expression of the *RNaseH1^{D210N}* catalytically dead mutant abolished S9.6 foci, thereby demonstrating the specificity of S9.6 antibody (Reviewer Figs. 6A, 6B). This figure is now included as Supp. Figs. 5C, 5D in the manuscript.

Reviewer Fig. 6. (A). Representative IF-FISH showing the specificity of S9.6 antibody (green) recognizing R-loops at UTs in U2OS cells expressing *RNaseH1^{WT}* or a catalytic dead *RNaseH1^{D210N}* mutant. UTs were detected with a Cy3-OO-(CCCTAA)₄ PNA probe (red) and 4,6-diamidino-2-phenylindole (DAPI) was used to stain nuclei (blue). (B). Quantification of S9.6 staining in (A). Data shown as the average of two independent experiments ± SD from a minimum 250 nuclei analyzed per experiment. *P=0.02 by one-way ANOVA. ns: non-significant.

The roles of DDX21 and ADAR1 in repressing ultrabright telomere formation appear clear from the data. However, a role for the TRF2 interaction in facilitating their function is not investigated. I suggest the authors disrupt the TRF2-DDX21 and/or TRF2-ADAR1 interactions with targeted mutations to more directly test the role of TRF2 in facilitating DDX21 and ADAR1 function at telomeres.

We present new data showing that both DDX21 and ADAR1p110 interact with the basic domain of TRF2. We postulate that TRF2^B recruits both proteins to telomeres to repress RNA/DNA hybrid and TERRA formation.

New co-IP experiments show that TRF2^B specifically mediate the interaction between TRF2 and DDX21 (Reviewer Fig. 7A). Co-IP of WT TRF2 and various TRF2 deletion mutants with DDX21 show that removal of TRF2^B severely disrupted interaction with DDX21. Deletion of both the basic and the Myb domains of TRF2 (but not the Myb domain by itself) completely eliminated interaction with DDX21. In a converse experiment, we show that the DDX21⁵⁸⁴⁻⁷⁸³ C-terminal domain is important for WT TRF2 interaction (Reviewer Fig. 7B). Both WT DDX21 and DDX21⁵⁸⁴⁻⁷⁸³ co-IPed with WT TRF2, while DDX21¹⁻⁵⁸³ interacted poorly with TRF2.

We found that the TRF2^B also mediates the interaction between TRF2 and ADAR1p110 (Reviewer Figs. 7C, 7D). Compared to TRF2^{AB}, WT TRF2 interacts three times stronger with WT ADAR1p110 (Reviewer Fig. 7C). Interaction between WT TRF2 and WT ADAR1p110 is independent of ADARp110's catalytic activity, since the catalytically dead ADAR1p110 E912A mutant interacted with WT TRF2 in a manner similar to WT ADAR1p110 (Reviewer Fig. 7D). These data suggest that direct interaction between DDX21 and ADAR1 is required to repress R-loops and UT formation. These figures are included as Figs. 5G, Supp. Figs. 5K-5M in the manuscript.

A

B

C

D

Reviewer Fig 7. (A) . Co-IP experiments showing that the basic domain of TRF2 is essential to interact with DDX21 in 293T cells. **(B)** . Co-IP showing the interaction between TRF2 and DDX21 mutant depends on the DDX21 C-terminus. **(C)** Co-IP experiments showing the interactions of TRF2^{WT} and TRF2^{ΔB} with GFP-ADAR1p110^{WT} in U2OS cells. **(D)** . Co-IP showing the interaction of TRF2 with Flag-ADAR1p110 is independent of ADAR1's catalytic activity.

Gaps in lamin A signal intensity at the nuclear envelope do not necessarily demonstrate a loss of compartmentalization as suggested by the authors (Fig. S6C,D). The authors should support this analysis by live-cell, time-lapse imaging of NLS-GFP reporter (PMID: 22567193). Loss of this marker is required to demonstrate a loss of compartmentalization. Other commonly used surrogate for nuclear envelope rupturing are localization of the DNA-binding proteins BAF and cGAS to the size of nuclear envelope rupture.

We thank the reviewer for this suggestion. Using GFP-cGAS as a marker for NE rupture as suggested by the reviewer, we provide new data showing increased GFP-cGAS staining at sites of nuclear envelope rupture in *Rap1*^{-/-} MEFs expressing TRF2^{ΔB} (Reviewer Figs. 8A-8C). ~25% of GFP-cGAS staining co-localized with telomeres. This data is now included as Supp. Figs. 6E, 6F in the manuscript.

A

B

Reviewer Fig 8. (A). Representative IF-FISH from *Rap1*^{-/-} cells showing that altered lamin A (blue) promotes GFP-cGAS (green) localization to nuclear envelope rupture sites. cGAS localization at nuclear envelope rupture sites with (red arrowhead) and without (white arrowhead) telomere co-localization. (B). Quantification of GFP-cGAS at nuclear envelope rupture sites in (A). Data shown as the average of three independent experiments \pm SD from a minimum 150 nuclei analyzed per experiment.

The authors focus a large portion of Fig. 7 on understanding how ATR-mediated phosphorylation affects Lamin A function. Much of this is based on unpublished data showing that ATR-mediated phosphorylation of Lamin A promotes micronucleus rupturing. Without the unpublished data it is very difficult to understand the rationale for promoting these experiments. I suggest the authors supply the unpublished data, cite the appropriate reference when appropriate, or remove this portion of Fig. 7. Related to the point above: It is not clear why ATR phosphorylates Lamin A.

Our new data eliminated the need to cite the unpublished data, and we have removed this reference. We provide new data to show that in addition to cGAS, we also found p-RPA32(S33) localized to NE rupture sites in *Rap1*^{-/-} MEFs expressing TRF2^{ΔB}, suggesting involvement of the ATR DNA damage pathway (Reviewer Figs. 9A, 9B). Since ATR is known to maintain NE integrity from mechanical or replication stress^{4,5}, we hypothesized that ATR-CHK1 activation in the absence of TRF2^B and RAP1 modulate NE and lamin A function. Since lamin A^{S395} is the predicted ATR phosphorylation site (using phospho ELM, a database of S/T/Y phosphorylation sites), we sought to determine whether lamin A^{S395} is an ATR substrate. We also postulate that lamin A^{S395} phosphorylation by ATR promotes lamin A disassembly, leading to increased inter-telomere homology search and telomere pairing during HDR. To test this hypothesis, we treated U2OS cells expressing lamin A phosphomimetic S395D or the phosphorylation deficient S395A mutants in U2OS cells reconstituted with vector or TRF2^{ΔB:L288R} with 4 different ATR inhibitors: AZD6738, AZD20, VE822 and BAY1895344. Both WT GFP-lamin A and the phosphorylation deficient lamin A^{S395A} localized predominantly to the nuclear lamina, resulting in a ~3-fold reduction of UTs (Figs. 7G-7I). In contrast, in cells expressing the phosphomimetic lamin A^{S395D}, we detected intense punctate nucleoplasm staining without any reduction in the number of UTs. Compared to DMSO treated cells, ATR inhibition greatly reduced lamin A^{S395D}-positive punctate nucleoplasm staining, accompanied by a significant reduction in the formation of UTs (Fig. 7G, Supp. Fig. 7B). In addition, telomere bridging, and HDR-induced telomere deletion were completely abolished by ATR inhibitors in *Rap1*^{-/-} MEFs expressing TRF2^{ΔB} (Supp. Figs. 7C-7E). These results suggest that ATR-mediated phosphorylation of lamin A^{S395} is required for the formation of UTs. Reviewer Figure 9 is now included as Supp. Figs. 6G, 6H in the manuscript.

Reviewer Fig 9. (A). Representative IF-FISH showing that p-RPA32 (S33) foci (green) localized to a NE rupture site (*) contain telomere (red arrowhead) in *Rap1*^{-/-} MEFs expressing the indicated proteins. p-RPA32 (S33) foci also colocalized with UTs (white arrowheads). (B). Quantification of NE rupture sites containing p-RPA32 (S33) foci with or without telomere in (A). Data shown as the average of three independent experiments \pm SD from a minimum 200 nuclei analyzed per experiment.

Reviewer #3 (Remarks to the Author):

We thank this reviewer for stating that “the presented data are compelling and support the claims of the authors. The findings are numerous and novel...in conclusion, I believe that the data presented here significantly advance our understanding of how shelterin protects telomeres and are of broad interest to the readers of Nature Communications.”

Many of the findings here would benefit from a more in-depth mechanistic analysis, which I understand is not feasible due to the large number of molecular processes analyzed here. Yet, a more compelling final model summarizing all the findings should be included.

We have now included a comprehensive figure as Reviewer Fig. 14 (Supp. Fig 8 in the manuscript).

In general, the observed phenotypes are reminiscent of ALT features. Unfortunately, the experiments are either performed in ALT cells or in MEFs, in which telomeres are very long and display singular chromatin. Therefore, some key finding must be reproduced in non-ALT human cells: colocalization of UTs with PML, SLX4, SMARCAL1.

We agree with the reviewer’s concerns. In our new data, we show that PML, SLX4, SMARCAL1 and p-RPA32 (S33) foci all co-localized to UTs in ALT-negative human HeLa cells (Reviewer Figs. 10A, 10B). We include Reviewer Fig. 10B as Sup. Fig. 4A in the manuscript.

A

B

Reviewer Fig. 10. (A). IF-FISH showing UTs colocalized with p-RPA32 (S33), PML, SLX4 and SMARCAL1 foci in HeLa cells. White arrowheads point to telomeres co-localized to the indicated proteins. (B). Quantification of percent UTs colocalized to p-RPA32, PML, SLX4 and SMARCAL1 in (A). Two independent experiments are shown as \pm SD from a minimum 150 nuclei analyzed per experiment. ****P<0.0001 by one-way ANOVA

To which scale are the phenotype observed ALT-like? Authors should provide C-circle and MIDAS analysis in non-ALT cells.

We observed UTs in both ALT-positive U2OS and telomerase-positive *Rap1*^{-/-} MEFs. We now provide new data showing ~3-fold increase in the amount of C-circles observed in *Rap1*^{-/-} MEFs expressing TRF2^{ΔB} as compared to vector control (Reviewer Fig. 1, Reviewer Fig. 11A). The amount of C-circles observed in *Rap1*^{-/-} MEFs is ~2.5-fold less than U2OS cells (Fig. 11B). In addition, both ALT-positive and ALT-negative cells are known to promote mitotic DNA synthesis (MiDAS) at telomeres^{6,7}. We asked whether UT formation due to loss of TRF2^B and RAP1 induce MiDAS. We observed EdU foci at single or both chromatids in *Rap1*^{-/-} MEFs expressing TRF2^{ΔB}, suggesting that TRF2^B and RAP1 normally suppress replication stress induced MiDAS (Reviewer Figs. 11C, 11D). Together, our data reveal that TRF2^B and RAP1 repress ALT-specific phenotypes and the localization of ALT-specific proteins to telomeres. This data is now included as Supp. Figs. 3I-3L in the manuscript.

Reviewer Fig. 11. (A). C-circle amplification from 400 μ g genomic DNA isolated from *Rap1*^{-/-} MEFs expressing indicated DNAs in the absence and presence Phi29 DNA polymerase, using ³²P-labeled-(CCCTAA)₄ telomeric probes. 400 μ g genomic DNA from U2OS cells was used as positive control for C-circle amplification. (B). Quantification of Phi 29 dependent C-circles in (A). The level of ³²P incorporation in the Phi29 negative control samples was subtracted from the samples that contained the Phi29 DNA polymerase. (C). Representative metaphase image showing MiDAS in *Rap1*^{-/-} MEFs expressing TRF2 ^{Δ B}. EdU labelled telomeres (green) were detected with a Cy3-OO-(CCCTAA)₄ PNA probe (red) and 4,6-diamidino-2-phenylindole (DAPI) was used to stain nuclei (blue). EdU signal in a single (white arrowhead) or both chromatids (orange arrowhead) (D). Quantification of percent EdU positive telomeres in (C). Data presents the average of two experiments \pm SD from 30 metaphase with each sample analyzed per experiment. *P=0.0274 by T test.

Does re-expression of ATRX reduce UTs in ALT cells?

We present new data to show that overexpression of ATRX reduced the number of UTs in U2OS cells (Reviewer Fig. 12A-C). Reviewer Figs. 12A, 12C are now included as Supp. Figs. 3B, 3C in the manuscript.

Reviewer Fig 12. (A). Immunoblot showing the reconstitution of GFP-ATRX in U2OS cells. Anti-GFP and anti-Myc antibodies were used to detect ATRX and Myc-tagged TRF2 ^{Δ B;L288R}. γ -Tubulin served as loading control. (B) IF-FISH showing the reconstitution of GFP-ATRX (green) in U2OS cells reduced the number of UTs (red). White arrowheads points to UTs. (C). Quantification of UTs in (B). Data shown as the mean of two experiments with \pm SD from a minimum 250 nuclei analyzed per experiment. *P=0.016, **P=0.005 by one-way ANOVA.

Is TERRA increased or just R-loops?

We present new data to show that both TERRA and R-loops are increased in U2OS cells expressing TRF2 ^{Δ B;L288R}. We show that accumulation of R-loops correlates with high TERRA levels at UTs (Reviewer Figs. 13A, 13B). This data is included as Supp. Figs. 5E, 5F in the manuscript.

A

B

Reviewer Fig. 13. (A). IF-FISH showing the expression of TERRA (green) in the presence of WT RNaseH or the catalytically dead RNaseH1^{D210N} mutant in U2OS cells expressing the indicated proteins. Telomeres (green) were detected with a Cy3-OO-(CCCTAA)₄ PNA probe (red) and 4,6-diamidino-2-phenylindole (DAPI) was used to stain nuclei (blue). (B). Quantification of TERRA in (A). Two independent experiments shown as \pm SD from a minimum 200 nuclei analyzed per experiment. *P=0.027

The Ku70/80 phenotype is puzzling. If RAP1-mediated recruitment Ku70/80 at telomeres mediates the Ku-dependent inhibition of HRD, then suppression of Ku in Rap1^{-/-} or TRF2-L286R cells should not have any effect. The data suggest a redundant recruitment of Ku, for which TRF2 have been suggested to play a role. To what extent is Ku70/80 lost at telomeres upon Rap1 suppression? This point should be at least discussed.

Loss of Rap1 does not impact Ku70/80 dependent C-NHEJ mediated telomere repair, suggesting that Ku70/80 is functioning normally in Rap1^{-/-} MEFs and not lost at telomeres. Both Rap1 and Ku70/Ku80 individually has been shown to suppress HDR at chromosome ends^{8,9,10}. Thus, expression of TRF2^{ΔB;L286R} in Ku70^{-/-}; Ku80^{-/-} MEFs resulted in additional increase in the number of telomere bridging and UTs observed as compared to Rap1^{-/-} MEFs expressing TRF2^{ΔB}. We discussed this point further in the manuscript.

As stated above, a compressive model including all the findings is needed to grasp and synthesize all the results presented here. The current model mostly focuses on Lamin A.

We agree with the reviewer and now provide a new compressive model to better illustrate the interplay of shelterin, telomeres and the nuclear envelope (Reviewer Fig. 14). This figure is now included as Supp. Fig. 8 in the manuscript.

Reviewer Fig. 14. Schematic illustrating the formation of ultrabright telomeres in the absence of RAPI and TRF2^{ΔB}. **A.** In WT cells, TRF2^B and RAPI cooperate to protect chromosome ends from inappropriate activation of ATR-dependent HDR. TRF2^B recruits DDX21 and ADAR1p110 to telomeres to resolve DNA-RNA hybrids, preventing the formation of telomeric R-loops, TERRA and UTs. In the absence of RAPI and TRF2^B, stalled replication forks lead to activation of ATR-CHK1, accumulation of telomeric R-loops and TERRA, promoting UT formation. Unresolved telomere recombination intermediates become substrates for cleavage by the SLX4 endonuclease, leading to catastrophic telomere shortening via T-loop HDR and the formation of telomere-free chromosome fusions. **B.** Lamin A acts as a scaffold that restrains telomere movement in the nucleus, possibly through transient associations of RAPI with Ku70-Ku80 and SUN1, known NE binding proteins. In the absence of RAPI and TRF2^B, ATR-CHK1 and CDK1 dependent phosphorylation of lamin A results in the disassembly of nuclear lamin A. Decompartamentalization of telomeres due to disruption of the lamin A architecture facilitates HDR, telomere-telomere clustering and UT formation.

In several instances, the text description does not correctly reflect the results. This should be corrected. In page 5, the authors state that UT formation “declined by 120h”. This is true for fig 1C but not Supp 1B-C. The text should be more specific.

We thank the reviewer for pointing out this error. Our time course experiments revealed that the number of UTs formed peaked at ~96h and declined by 120h (Fig. 1C). However, the decline in the number of UTs correlates with an increase in its size due to increase in HDR (Supp. Fig. 1B, 1C). We have modified this point in the text.

In page 6, it is stated that depletion of RAD51 decreased formation of UTs by 3-fold, yet there is no quantification.

Fig. 2b show shRAD51 quantification (purple bar).

Twice in page 6, in pages 8 and 9, it is question of “100%” co-localization, which is not correct. Replace by “nearly 100%”.

We thank the reviewer for pointing out this error. We have corrected the text to nearly 100% at page 6, 8 and 9.

In page 7, it is stated that TRF1-FOK1 results in an increase in average telomere size in U2OS and MEFs. However, in MEFs, the differences are not statistically significant.

We have corrected this statement.

References

1. Tumini E, Barroso S, Calero CP, Aguilera A. Roles of human POLD1 and POLD3 in genome stability. *Sci Rep* **6**, 38873 (2016).
2. Layer JV, *et al.* Polymerase delta promotes chromosomal rearrangements and imprecise double-strand break repair. *Proc Natl Acad Sci U S A* **117**, 27566-27577 (2020).
3. Liao H, *et al.* CDK1 promotes nascent DNA synthesis and induces resistance of cancer cells to DNA-damaging therapeutic agents. *Oncotarget* **8**, 90662-90673 (2017).
4. Kidiyoor GR, *et al.* ATR is essential for preservation of cell mechanics and nuclear integrity during interstitial migration. *Nat Commun* **11**, 4828 (2020).
5. Shah P, *et al.* ATM Modulates Nuclear Mechanics by Regulating Lamin A Levels. *Front Cell Dev Biol* **10**, 875132 (2022).
6. Min J, Wright WE, Shay JW. Alternative Lengthening of Telomeres Mediated by Mitotic DNA Synthesis Engages Break-Induced Replication Processes. *Mol Cell Biol* **37**, (2017).
7. Ozer O, Bhowmick R, Liu Y, Hickson ID. Human cancer cells utilize mitotic DNA synthesis to resist replication stress at telomeres regardless of their telomere maintenance mechanism. *Oncotarget* **9**, 15836-15846 (2018).
8. Sfeir A, Kabir S, van Overbeek M, Celli GB, de Lange T. Loss of Rap1 induces telomere recombination in the absence of NHEJ or a DNA damage signal. *Science* **327**, 1657-1661 (2010).
9. Chen Y, *et al.* A conserved motif within RAP1 has diversified roles in telomere protection and regulation in different organisms. *Nat Struct Mol Biol* **18**, 213-221 (2011).
10. Rai R, Chen Y, Lei M, Chang S. TRF2-RAP1 is required to protect telomeres from engaging in homologous recombination-mediated deletions and fusions. *Nat Commun* **7**, 10881 (2016).

REVIEWER COMMENTS

Reviewer #1 (Remarks to the Author):

Overall, the authors have done a good job addressing the comments, and I believe this has improved the manuscript. I still think that it is difficult to connect the different aspects of the manuscript. This could be improved in the discussion and by including the model (currently included as a Supp figure) as a main figure.

I do not agree with using MEFs as the telomerase-positive cell line and making conclusions based on this. Mouse telomeres are different, they are very long, have some level of instability (including C-circles), and can undergo recombination. I do not think that MEFs should be used as the telomerase-positive system when comparing with U2OS. This should be a human telomerase-positive cell line (eg HeLa). Alternatively, this caveat should be articulated clearly in the manuscript. I would actually be surprised if you see the same in a human telomerase-positive cell line (no images are provided for the HeLa data in Supp 4A, and the columns on this figure are not defined). This also raises the question why a human telomerase-positive cell line was not used in the first place. This outstanding point should be addressed.

Reviewer #2 (Remarks to the Author):

The authors have supplied a point by point response to address my prior concerns regarding their manuscript. As detailed below there are several examples that were completely addressed however, other points were not satisfactorily addressed and I raise several points of concern regarding the newly included data. Overall - despite the abundance of data and clearly very hard work performed here - I remain unconvinced about central aspects of the model.

The point regarding potential artifacts associated with use of the S9.6 antibody was mostly addressed. One small point of confusion: I had assumed the S9.6 signal in the cytoplasm was an artifact - yet this is reduced by overexpression of RNaseH1-wt (not D210N). It would be helpful if the authors could elaborate on potential sources of cytoplasmic S9.6 signal.

The point regarding directly testing how TRF2-DDX21, TRF2-ADAR1 interactions facilitate function was not satisfactorily addressed. The authors obtain relatively precise mutants of TRF2 that disrupt the interaction but do not establish functional significance. TRF2 basic domain is involved in many other aspects of telomere function therefore the specificity of these mutations is unknown.

I am concerned about the GFP-cGAS imaging data shown in Rev Fig. 8A. The GFP-cGAS foci shown do not resemble any published examples of GFP-cGAS at nuclear rupture sites. Please see PMID: 30171044, 34551315, 27013428, 30811988 and many other examples in the literature. I am concerned that these images may be artifacts.

The point regarding the role of ATR in regulating Lamin A function was partly addressed. The authors have removed unsupported statements as requested but the new data included do not strongly support the claims made. In particular, detection of p-RPA32(S33) foci at NE rupture sites do not necessarily indicate that ATR is involved. The pRPA foci shown in Reviewer Fig. 9A seem to be spread throughout the nucleus. The quantification shown in Reviewer Fig. 9B measures pRPA32 foci at nuclear rupture sites yet there are no IF data shown marking rupture sites. Therefore it is not clear how the authors identified rupture sites in these experiments. Finally, there is no strong evidence that ATR directly phosphorylates lamins.

Reviewer #3 (Remarks to the Author):

In this reviewed manuscript, Rekha Rai, Sandy Chang and colleagues investigate the role the TRF2-basic domain and Rap1 in preventing HDR at telomeres. The data are novel and significant to understand the molecular mechanisms of telomere protection.

I had few questions/concerns on the first submission, and they were all rigorously addressed in this revised version. I am favorable to publication of the current manuscript in Nature Communications

Second Round of Reviewers' Comments

Reviewer #1 (Remarks to the Author):

Overall, the authors have done a good job addressing the comments, and I believe this has improved the manuscript. I still think that it is difficult to connect the different aspects of the manuscript. This could be improved in the discussion and by including the model (currently included as a Supp figure) as a main figure.

We thank this reviewer and have now modified the discussion and included the schematic as Fig. 8.

I do not agree with using MEFs as the telomerase-positive cell line and making conclusions based on this. Mouse telomeres are different, they are very long, have some level of instability (including C-circles), and can undergo recombination. I do not think that MEFs should be used as the telomerase-positive system when comparing with U2OS.

We agree with the reviewer that mouse telomeres are longer than human cells. However, we never observed C-circles in vector or shTrf2 treated cells. C-circles were observed only in Rap1^{-/-} MEFs expressing TRF2^{ΔB} suggesting that TRF2^B and RAP1 repress ALT-specific phenotypes (Reviewer Fig. 1). These figures are included in the manuscript as Supp. Fig. 3I, 3J.

A

B

Reviewer Fig. 1. (A). C-circle amplification from 400μg of genomic DNA isolated from Rap1^{-/-} MEFs expressing indicated DNAs in the absence and presence Phi29 DNA polymerase, using ³²P-labeled-(CCCTAA)₄ telomeric probes. 400μg genomic DNA from U2OS cells was used as positive control for C-circle amplification. **(B).** Quantification of Phi 29 dependent C-circles in (A). For quantification, the level of ³²P incorporation in the Phi29 negative control samples was subtracted from the samples that contained the Phi29 DNA polymerase.

This should be a human telomerase-positive cell line (eg HeLa). Alternatively, this caveat should be articulated clearly in the manuscript. I would actually be surprised if you see the same in a human telomerase-positive cell line (no images are provided for the HeLa data in Supp 4A, and the columns on

this figure are not defined). This also raises the question why a human telomerase-positive cell line was not used in the first place. This outstanding point should be addressed.

We thank the reviewer for pointing out undefined columns in Supp. Fig. 4A. The columns on this figure are defined now.

We already present data on UT formation in the human cell line IMR90 (Supp. Fig. 1A). In our first rebuttal, we presented data on UT formation in HeLa cells. We apologize for not including their images in Supp Fig. 4A in the manuscript due to space limitation. However, these images and quantification for UTs in HeLa cells were presented in our first rebuttal as Reviewer Fig. 10A. We now provide Reviewer Fig. 2 to show UTs images in HeLa cells. This data is now incorporated in the manuscript as Supp. Fig. 4A, 4B.

A

B

Reviewer Fig. 2: (A). IF-FISH showing UTs colocalized with p-RPA32 (S33), PML, SLX4 and SMARCAL1 foci in HeLa cells. White arrowheads point to UTs co-localized to the indicated proteins. (B). Quantification of percent UTs colocalized to p-RPA32, PML, SLX4 and SMARCAL1 in (A). Two independent experiments are shown as \pm SD from a minimum 150 nuclei analyzed per experiment. ****P<0.0001 by one-way ANOVA

Reviewer #2 (Remarks to the Author):

The authors have supplied a point by point response to address my prior concerns regarding their manuscript. As detailed below there are several examples that were completely addressed however, other points were not satisfactorily addressed and I raise several points of concern regarding the newly included data. Overall - despite the abundance of data and clearly very hard work performed here - I remain unconvinced about central aspects of the model.

The point regarding potential artifacts associated with use of the S9.6 antibody was mostly addressed. One small point of confusion: I had assumed the S9.6 signal in the cytoplasm was an artifact - yet this is reduced by overexpression of RNaseH1-wt (not D210N). It would be helpful if the authors could elaborate on potential sources of cytoplasmic S9.6 signal.

We thank the reviewer for pointing out the cytoplasmic S9.6 staining in our images. This cytoplasmic staining is known background staining from the S9.6 antibody. We now provide images showing this cytoplasmic staining in all of our IF panels (Reviewer Fig. 3). This figure is now included as Supp. Fig. 5C in the manuscript.

Reviewer Fig. 3. Representative IF-FISH showing the specificity of S9.6 antibody (green) recognizing R-loops at UTs in U2OS cells expressing RNaseH1^{WT} or a catalytic dead RNaseH1^{D210N} mutant. UTs were detected with a Cy3-OO-(CCCTAA)₄ PNA probe (red) and 4,6-diamidino-2-phenylindole (DAPI) was used to stain nuclei (blue).

The point regarding directly testing how TRF2-DDX21, TRF2-ADAR1 interactions facilitate function was not satisfactorily addressed. The authors obtain relatively precise mutants of TRF2 that disrupt the interaction but do not establish functional significance. TRF2 basic domain is involved in many other aspects of telomere function therefore the specificity of these mutations is unknown.

We have already provided data showing that the basic domain of TRF2 is required for functional interaction with DDX21 and ADAR1 (Figures 5G-I). We are currently mapping the residues within the TRF2 basic domain that is responsible for these interactions, but strongly feel that presenting this data is beyond the scope of this manuscript.

I am concerned about the GFP-cGAS imaging data shown in Rev Fig. 8A. The GFP-cGAS foci shown do not resemble any published examples of GFP-cGAS at nuclear rupture sites. Please see PMID: 30171044, 34551315, 27013428, 30811988 and many other examples in the literature. I am concerned that these images may be artifacts.

We strongly disagree with this reviewer that our GFP-cGAS foci are artifacts. The pMSCVpuro-eGFP-mcGAS plasmid used in our study was obtained from Mackenzie et al. (<https://pubmed.ncbi.nlm.nih.gov/28738408/>) and it showed an almost identical staining pattern as images displayed in the Mackenzie paper, indicating that cGAS foci observed in our experimental setting are not artifacts (Reviewer Fig. 4A). We now provide new data showing that GFP-cGAS reconstituted in $cGas^{-/-}$ MEFs localized to micronuclei upon NE rupture after 1 Gy irradiation (Reviewer Fig. 4B). The punctate staining in this positive control is identical to those found in the Mackenzie et al paper, as well as those found in our experimental settings (compare Reviewer Fig. 4A with Fig. 4B). We now include Reviewer Fig. 4B as Supp. Fig. 6E in the manuscript.

A.

B
Reviewer Fig. 4. *A. Representative IF-FISH from $Rap1^{-/-}$ cells showing that altered lamin A (blue) promotes GFP-cGAS (green) localization to nuclear envelope rupture sites. cGAS localization at nuclear envelope rupture sites is shown with white arrowhead. B. GFP-cGAS DNA (green) reconstituted in $cGas^{-/-}$ MEFs localized to NE rupture site (white arrowhead) after 1 Gy IR. Lamin B1 (red) stains the nuclear envelope.*

The point regarding the role of ATR in regulating Lamin A function was partly addressed. The authors have removed unsupported statements as requested but the new data included do not strongly support the claims made. In particular, detection of p-RPA32(S33) foci at NE rupture sites do not necessarily indicate that ATR is involved. The pRPA foci shown in Reviewer Fig. 9A seem to be spread throughout the nucleus. The quantification shown in Reviewer Fig. 9B measures pRPA32 foci at nuclear rupture sites yet there are no IF data shown marking rupture sites. Therefore, it is not clear how the authors identified rupture sites in these experiments. Finally, there is no strong evidence that ATR directly phosphorylates lamins.

In addition to inducing p-RPA32 foci localization at NE rupture sites, expression of TRF2^{ΔB} in $Rap1^{-/-}$ MEFs leads to the localization of p-RPA32 to UTs, thus spreading p-RPA32 foci throughout the nucleus (Fig. 2A, Supp Fig. 2A). We now include data to show that NE rupture sites contain p-RPA32 (green, marked with an asterisk) (Reviewer Fig. 5A). The quantification shown in Reviewer Fig. 5B measures p-RPA32 foci only at ruptured NE (marked with an asterisk).

Recent literature suggests that NE rupture or structural alteration at the NE activates ATR, leading to the recruitment of DDR factors including p-RPA32:

Kidiyoor GR, et al. ATR is essential for preservation of cell mechanics and nuclear integrity during interstitial migration. Nat Commun 11, 4828 (2020).

Shah P, et al. Nuclear Deformation Causes DNA Damage by Increasing Replication Stress. Curr Biol 31, 753-765 e756 (2021).

Shah P, et al. ATM Modulates Nuclear Mechanics by Regulating Lamin A Levels. Front Cell Dev Biol 10, 875132 (2022).

These reports support our observation that p-RPA32 foci localize to NE rupture sites as shown in Reviewer Fig. 5C.

ATR inhibition greatly reduced lamin A^{S395D} -positive punctate nucleoplasm staining, accompanied by a significant reduction in the number of UTs (Fig. 7G, Supp. Fig. 7B). In addition, telomere bridging, and HDR-induced telomere deletion were completely abolished by ATR inhibitors in $Rap1^{-/-}$ MEFs expressing $TRF2^{\Delta B}$ (Supp. Fig. 7C-7E). These results strongly suggest that ATR-mediated phosphorylation of lamin A^{S395} is required for the formation of UTs. However, we do agree that these data do not provide definitive evidence that ATR directly phosphorylates lamins. We have modified the text to make this point clear.

A

B

C

Reviewer Fig. 5. (A). Representative IF-FISH showing that p-RPA32 (S33) foci (green) localized to a NE rupture site (*) contain telomere (red arrowhead) in $Rap1^{-/-}$ MEFs expressing the indicated proteins. p-RPA32 (S33) foci also colocalized with UTs (white arrowheads). (B). Quantification of NE rupture sites containing p-RPA32 (S33) foci with or without telomere in (A). Data shown as the average of three independent experiments \pm SD from a minimum 200 nuclei analyzed per experiment. (C). Multiple IF-FISH images showing p-RPA32 (S33) foci (green) localized to NE rupture site (*) containing telomere (red arrowhead) in $Rap1^{-/-}$ MEFs expressing the indicated proteins. p-RPA32 (S33) foci also colocalized with UTs (white arrowheads).

REVIEWER COMMENTS

Reviewer #1 (Remarks to the Author):

The authors have completed a second round of amendments to the manuscript to address the remaining concerns raised by reviewers 1 and 2. Specifically, the authors have included data from HeLa cells, further validation of the S9.6 Ab, validation of nuclear rupture sites/staining, and toned down conclusions regarding ATR signalling. I feel that the authors have done an adequate job addressing the remaining points with both data and amendments to the text. I support publication in Nat Commun.

Reviewer #4 (Remarks to the Author):

The authors found that Rap^{-/-} MEFs reconstituted with TRF2dB exhibited NE rupture phenotypes (Fig. 7A, B) as well as micronuclei enriched with GFP-cGAS. However, Reviewer 2 argued that the GFP-cGAS foci do not resemble the examples in several papers he mentioned. The authors rebutted this issue with positive control data showing the localization of GFP-cGAS to the IR-induced micronuclei. Based on our experiences, nuclear rupture induced by different mechanisms can result in micronuclei of distinct structures and GFP-cGAS foci. Since the examples of nuclear rupture provided by Reviewer 2 are associated with cell migration, cancer invasion, or artificially induced in a microfabricated cell confiner, it is not surprising that the GFP-cGAS foci do not look like those induced by TRF2dB in Rap^{-/-} MEFs. Thus, I actually agree with the authors' statements about GFP-cGAS foci. On the other hand, I concur with Reviewer 2 on the point that more evidence is required to prove that Lamin A is regulated by ATR to affect UT and nuclear rupture. Regarding pRPA32(S33) in nuclear ruptures, I think the Reviewer Figures 5a, b show nicely that pRPA32(S33) is localized at the nuclear ruptures.